# Annotation of natural product compound families using molecular networking topology and structural similarity fingerprinting

Nicholas J. Morehouse [1], Trevor N. Clark[2], Emily J. McMann[2], Jeffrey A. van Santen [2], F. P. Jake Haeckl [2], Christopher A. Gray[1,3] & Roger G. Linington [2] ✉

Spectral matching of MS[2] fragmentation spectra has become a popular method for characterizing natural products libraries but identification remains challenging due to differences in MS[2] fragmentation properties between instruments and the low coverage of current spectral reference libraries. To address this bottleneck we present Structural similarity Network Annotation Platform for Mass Spectrometry (SNAP-MS) which matches chemical similarity grouping in the Natural Products Atlas to grouping of mass spectrometry features from molecular networking. This approach assigns compound families to molecular networking subnetworks without the need for experimental or calculated reference spectra. We demonstrate SNAP-MS can accurately annotate subnetworks built from both reference spectra and an in-house microbial extract library, and correctly predict compound families from published molecular networks acquired on a range of MS instrumentation. Compound family annotations for the microbial extract library are validated by co-injection of standards or isolation and spectroscopic analysis. SNAP-MS is freely available at www.npatlas.org/discover/snapms.

The identification of molecules within complex mixtures is a major bottleneck in natural products research and untargeted metabolomics studies. Mass spectrometry has emerged as the de facto tool for the high-throughput characterization of metabolites because of its sensitivity compared to other methods. However, only limited structural information is contained in tandem mass spectrometry (MS[2]) data. Identifications can sometimes be made by comparing MS[2] spectra to spectral reference libraries, but the coverage of MS[2] reference spectra for natural products remains low[1–3]. In silico methods for calculating MS[2] spectra can alleviate this issue, but still struggle to accurately predict MS[2] spectra for many classes of natural products[4,5]. These issues persist, despite continuing improvements to prediction techniques[6,7], the recent development of new scoring algorithms to rank matches between experimental and calculated MS[2] spectra[8], and the development of tools, such as Network Annotation Prediction

(NAP) aimed at reranking the candidates provided by in silico predictions[9]. Additionally, MS[2] spectral networking can be used to generate molecular networks, which group features based on similarities in MS[2] spectra and can allow for propagation of annotations to unknown molecules when annotations are possible, but often return lists of potential candidates that must be manually examined by end users[10–12]. Thus, while MS[2] spectral networking can be used to group molecules by structural similarity, it is often not yet possible to identify these compound families.

A second often overlooked challenge for analyte identification by mass spectrometry is the diversity of data types available from 'high-resolution' mass spectrometers. Variations in acquisition conditions (e.g., source temperature, analyte concentration, cone voltage, mobile phase), experimental parameters (e.g., instrument polarity, data-dependent vs. data-independent MS[2] fragmentation) and instrument

[1]Department of Biological Sciences, University of New Brunswick, Saint John, NB, Canada. [2]Department of Chemistry, Simon Fraser University, Burnaby, BC, Canada. [3]Department of Chemistry, University of New Brunswick, Fredericton, NB, Canada. ✉e-mail: rliningt@sfu.ca

configuration (e.g., qTOF vs. Orbitrap) all combine to introduce significant variations in MS[2] spectra for the same analyte analyzed under different conditions[13]. In addition, some instruments are not even capable of acquiring MS[2] data, precluding the use of spectral matching. Recently our laboratory led the development of the Natural Products Atlas database to create a comprehensive collection of all published microbial natural products[14]. Interestingly, examination of the distribution and co-occurrence of molecular formulae within the Natural Products Atlas has revealed that natural products are not distributed randomly in chemical space, but instead are grouped closely around specific scaffolds[15]. Structural variation in each scaffold is therefore a unique fingerprint for compound class. We hypothesize that intra-family formula distributions can be used for the de novo identification of subnetworks defined by MS[2] spectral networking from tools such as molecular networking.

In this paper we present Structural similarity Network Annotation Platform for Mass Spectrometry (SNAP-MS), a tool for annotation of MS[2] spectral networking subnetworks based on formula distributions in microbial natural products libraries. Recognizing that many of the tools to be discussed in this study utilize similar language with nuanced meanings, we have provided a detailed glossary of terms to assist readers (Supplementary Table 1). This strategy exploits the observation that formula distributions are diagnostic of compound families to annotate groupings of mass spectral features in molecular networks directly from MS[1] data, without the need for experimental or calculated MS[2] reference libraries. First, we discuss the distributions of molecular formulae among compound families in the Natural Products Atlas. Second, we identify a compound similarity scoring method that mimics groupings derived from MS[2] spectral networking. Third, we present the underlying algorithms that drive compound family identification in SNAP-MS and evaluate performance using a molecular network built from pure compound reference spectra. Fourthly, we show that SNAP-MS can correctly annotate compound families from both in-house and published molecular networks. Analysis of a 925-member in-house microbial extract fraction library yielded annotations for 11 compound families, seven of which were confirmed by orthogonal spectroscopic methods. Separately, analysis of six published molecular networking subnetworks yielded annotations that matched the published compound classes in all cases. Overall, SNAP-MS predicted the correct compound class in 31 of the 35 annotated subnetworks; a success rate of 89%. Finally, we extend the SNAP-MS methodology to plant and invertebrate natural products chemistry using the COCONUT database[16], and compare the performance of SNAP-MS to Network Annotation Propagation, another open access annotation tool for annotation prediction. Together these results illustrate the value of SNAP-MS for the structural annotation of large untargeted metabolomics datasets without the need for experimentally acquired spectral libraries.

## Results

SNAP-MS exploits the observations that natural products chemical diversity is grouped around core scaffolds for many compound families, and that formula distributions within these compound families are almost always unique. Using this information, we hypothesized that formula distributions could be used to annotate groupings formed by MS[2] spectral networking in the absence of other analytical data (MS[n] fragmentation, clogP, NMR spectra etc.) by relating formula distribution to a compound family. To test this hypothesis, we first examined the distributions of chemical formulae between compound families in the Natural Products Atlas. Next, we identified a chemoinformatic compound clustering method that closely mirrored the subnetworks obtained from MS[2] spectral networking (molecular networking). Finally, we developed a set of algorithms to apply this method to subnetworks from molecular networking graphs.

## Molecular formula distributions are diagnostic for compound family

Using the Natural Products Atlas database (v2020_06) as a source of chemical structures, we examined the occurrence, distribution and grouping of molecular formulae in microbial natural products. The goal of this analysis was to test the hypothesis that intra-family formula distributions are diagnostic identifiers for natural product compound families. Within the 29,006 compounds in the dataset there were 12,666 unique molecular formulae, including 8349 formulae that appeared only once (Supplementary Fig. 1). We observed that the most common molecular formulae exclusively contain the elements carbon, hydrogen, and oxygen, with $C_{15}H_{22}O_3$ being the most common formula (151 instances), followed by $C_{15}H_{24}O_3$ (127 instances). This is perhaps not surprising, given that several of the major biosynthetic classes (e.g., polyketides, saccharides, terpenes) are constructed from building blocks that contain exclusively these three elements. In fact, the most common formulae that contain elements in addition to carbon, hydrogen, and oxygen ($C_{23}H_{31}NO_4$ and $C_{24}H_{35}NO_4$) are present in just 15 instances.

Grouping these molecules into compound families using the standard clustering method from the Natural Products Atlas (Morgan fingerprinting (radius = 2) and Dice similarity scoring (0.75 cutoff)) yielded 8381 compound families (https://doi.org/10.5281/zenodo.3981180), 1901 when excluding those with fewer than three members (Supplementary Fig. 2). Excluding formulae that appear only once in the Natural Products Atlas, we tested the hypothesis that formula distributions are indicative of compound family by counting the occurrence of each molecular formula, each pair of molecular formulae, and each set of three molecular formulae across compound families (Fig. 1, for a full analysis, including single formulae, see Supplementary Fig. 3). Of the remaining 4317 unique molecular formulae, 36% are present in only one compound family (Fig. 1a). In contrast to the moderate discriminatory power of individual formulae, sets of

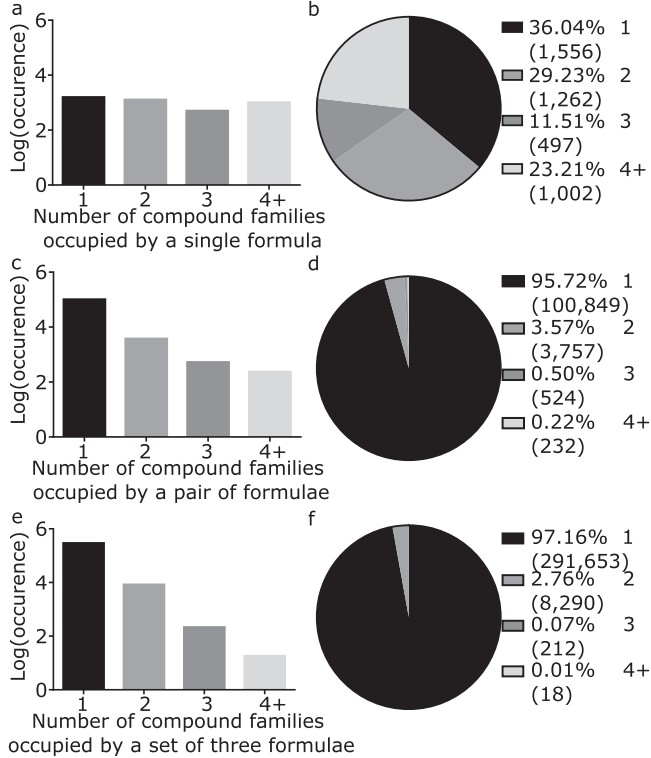

**Fig. 1 | Intra-family distributions of molecular formulae.** The log-transformed occurrence (bar charts) or proportion (pie charts) of **a**, **b** a single molecular formula, **c**, **d** a pair of molecular formulae, or **e**, **f** a set of three molecular formulae appearing in one or more compound families in the Natural Products Atlas.

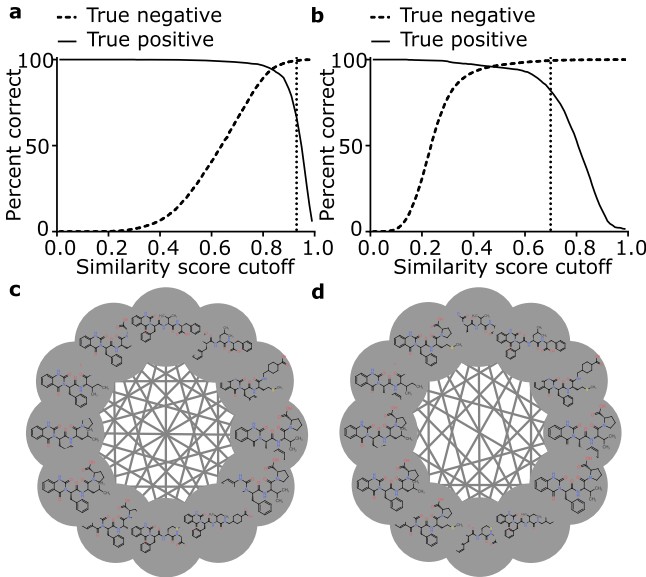

**Fig. 2 | Alignment of cheminformatic chemical fingerprint similarity scoring with MS² spectral networking.** Alignment between structural similarity networks generated using **a** MACCS fingerprinting with Dice scoring or **b** Morgan fingerprinting (Radius = 2) with Dice scoring, with a molecular network created from GNPS public spectral library GNPS-NIH-NATURALPRODUCTSLIBRARY. True positives are edges that exist between nodes in the molecular network while true negatives are edges which do not exist between nodes in the molecular network. Vertical dotted lines show the location where false connections equal 0.5%. **c** A subnetwork from the resulting GNPS network built from the NIH spectral libraries. **d** A structural similarity network built using the molecules present in panel **c** illustrating that the absence of selected edges does not substantively impact the interconnectivity of nodes within the graph, Morgan fingerprinting (Radius = 2) and minimum dice score of 0.71.

formulae were found to be highly diagnostic for compound family. Specifically, both pairs and sets of three molecular formulae were found to be present in single compound families in >95% and >97% of cases, respectively (Fig. 1c-f). Even the most common sets of three formulae belong to few compound families, despite the fact that each formula corresponds to many candidate molecules. For example, $C_{10}H_{10}O_4$, $C_{11}H_{12}O_4$, and $C_{11}H_{12}O_5$ correspond to 33, 39, and 39 molecules each, yet this formula combination occurs in only five compound families. These results demonstrate that formula distributions are indicative of compound family, even for very common formulae. Of particular note, there are almost no examples of sets of three formulae that appear in more than one compound family (<3%), making these sets highly diagnostic for compound structural class.

### Cheminformatic compound clustering aligns with compound clustering based on molecular networking

To correctly identify compound families using structural similarity, the compound families produced by cheminformatic chemical fingerprinting must align closely with the groupings from MS² spectral networking. To select the optimal theoretical clustering method, we compared subnetworks generated by molecular networking of 1267 spectra from known standards (Methods) against compound families generated using a panel of common chemical fingerprinting and similarity scoring methods using a range of similarity score cutoffs (Methods). Because incorrect connections between compound families (false positives) have a much greater impact on overall network structure than missing connections within families (false negatives) we required a clustering method that delivered the highest possible true positive rate with a very low false positive rate (<0.5%). To assess the overall influence of clustering method on network structure we plotted

the percentages of both true negative and true positive connections against similarity score cutoff (Fig. 2a, b). At a false positive rate of 0.5% MACCS keys clustering showed the lowest alignment with molecular networking. This combination of methods required a very high Dice similarity score (0.94) to reduce the false positive rate below 0.5%. At this cutoff only 58% of the expected true positive connections between structures were present, leading to the creation of a highly fragmented network with many small clusters that were not consistent with the MS-based network (Fig. 2a). By contrast, Morgan fingerprinting with a radius of 2, 4 or 6 using either Tanimoto or Dice scoring produced compound families with excellent alignment to molecular networking (Fig. 2b). We selected Morgan fingerprinting (radius = 2) and Dice similarity scoring (0.71 cutoff) as the cheminformatic compound clustering method with the best alignment to subnetworks generated by molecular networking (Fig. 2c, d). These results demonstrate that compound families generated by cheminformatic compound clustering align well with groupings derived from MS-based molecular networking. Based on this result we developed a scoring platform, SNAP-MS, for the de novo identification of molecular networking subnetworks using formula distributions.

### Compound family identification using SNAP-MS

The SNAP-MS workflow annotates molecular networking subnetworks using the following four steps: import cluster MS data, extract candidate matches for each cluster mass from the reference database, group candidate matches using the compound clustering tool, and finally filter and prioritize results groups based on subnetwork coverage (Fig. 3). To illustrate this workflow, we examined one subnetwork containing five masses from our marine bacteria library (Fig. 3a). Searching each mass against the Natural Products Atlas database considering a panel of possible counterions ([M + H]⁺, [M + Na]⁺, [M-$H_2O$ + H]⁺) yielded a total of 17 candidate molecules (Fig. 3b). Candidate molecules were grouped into compound families using the similarity metric and cutoff described above, resulting in 11 compound families. These results were then filtered to remove small families (≤2 members), leaving only one candidate compound family (6 members). The central premise of SNAP-MS is that, because formula distributions are diagnostic of compound family, the correct assignment will be the family with the highest number of matches to masses from the original MS² cluster. Therefore, even though each mass generates a large pool of candidate structures, most candidates will be eliminated because they are not structurally related to other candidates. In this example, SNAP-MS clustered 17 candidate molecules into 11 compound families. Because some compound families contain large numbers of isobaric regio- and stereoisomers it is not appropriate to rank compound families by cluster size. Instead, ranking is based on how many masses from the original subnetwork are accounted for by molecules within each candidate compound family. In this case, one compound family occupied the top-ranked position with structures that matched three masses in the original subnetwork (Fig. 3c), suggesting that this subnetwork contains members of the trichostatin compound family. Subsequent isolation and characterization of trichostatin A by NMR spectroscopy confirmed the annotation, validating the SNAP-MS approach for compound family annotation (Fig. 3d).

### SNAP-MS correctly annotates compound networks for pure compound reference spectra

To assess the accuracy of SNAP-MS we first analyzed subnetworks from a molecular network built from the GNPS public spectral libraries NIH Natural Products Library 1 and NIH Natural Products Library 2 (Methods). We selected these reference libraries because they were publicly available, contained MS² data for individual pure compounds, and overlapped with compounds in the Natural Products Atlas (Fig. 4). The 9182 reference spectra were submitted to GNPS for molecular networking analysis, and the resulting network filtered to remove subnetworks that contained fewer than three compounds from the

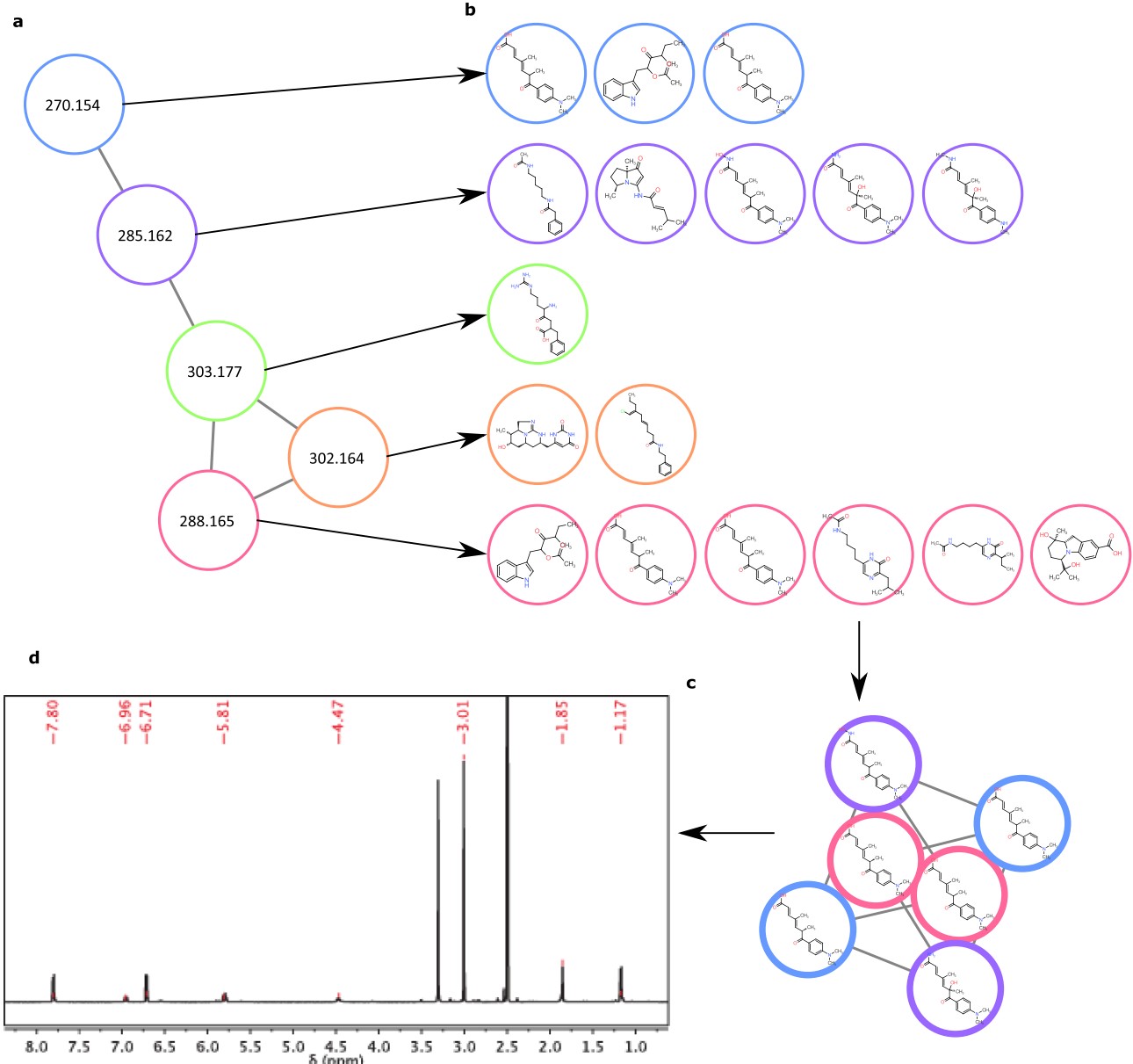

**Fig. 3 | SNAP-MS workflow.** The compound family annotation of a subnetwork showing **a** the input subnetwork, **b** candidate structures retrieved from the Natural Products Atlas for each mass from panel a. Node edge color indicates original mass match, **c** the resulting top ranked compound family after structural similarity scoring and filtering containing members of the trichostatin compound family, and **d** confirmation of the assignment by isolation and ¹H NMR analysis of trichostatin A.

Natural Products Atlas. The resulting subset contained 22 subnetworks with a total of 306 nodes, of which 139 corresponded to molecules present in Natural Products Atlas (Fig. 4). SNAP-MS analysis provided compound family identifications for all but two of these subnetworks. The correct answer was ranked first 17 times, ranked second once, and incorrectly predicted twice. However, review of the subnetworks that were incorrectly assigned revealed that they contained compounds from structurally dissimilar compound families (Supplementary Fig. 4). This result highlights the influence that MS² input data quality has on grouping methods such as molecular networking, particularly for annotation tools such as SNAP-MS that rely on these groupings to make compound family predictions. Excluding the mis-assembled subnetworks, SNAP-MS returned the correct compound family in 18 of 20 cases: a success rate of 90%.

Figure 4 demonstrates that SNAP-MS can correctly identify compound families from MS² spectral networking even when the input data contains nodes that are not part of the reference database, and that

these identifications do not require experimental reference spectra. For example, the subnetwork analyzed in section B contains 15 members of the diphenyl ether family. Six of these compounds are present in the Natural Products Atlas dataset (green nodes) with a further nine not currently included (grey nodes). Mass matching against the Natural Products Atlas returned candidate molecules for all nodes, with an average of 3.1 candidates per node. However, as expected, structural similarity among incorrect candidates was low, and these molecules did not form compound families with broad subnetwork coverage. Overall, the 30 initial candidate molecules formed just four compound families and of these only one, the correct one, accounted for more than two of the subnetwork nodes.

## Identification of compound families from marine bacterial extracts
To evaluate the performance of this method for complex mixtures we next analyzed a molecular network built from 925 samples from our in-

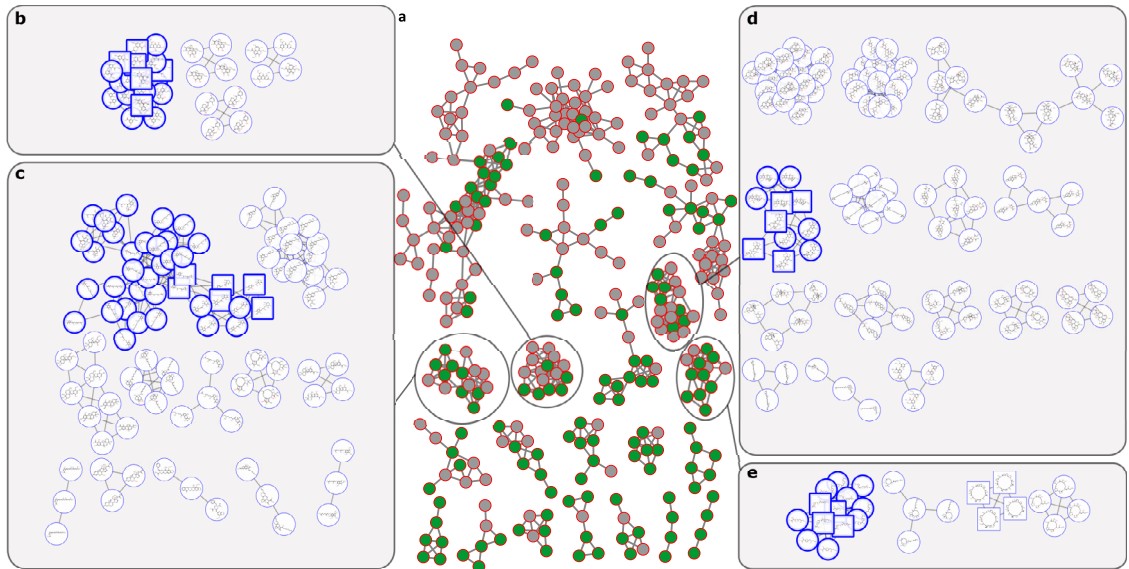

**Fig. 4 | Evaluation of SNAP-MS using a molecular network built from reference spectra. a** Subnetworks from a molecular network were created from GNPS public spectral libraries NIH Natural Products Library 1 and NIH Natural Products Library 2. Nodes corresponding to compounds present in the Natural Products Atlas colored green. **b–e** Predicted compound families for select subnetworks. Molecules present in the original molecular networking subnetwork are indicated with square nodes. The top ranked answers from SNAP-MS are indicated with a bold border.

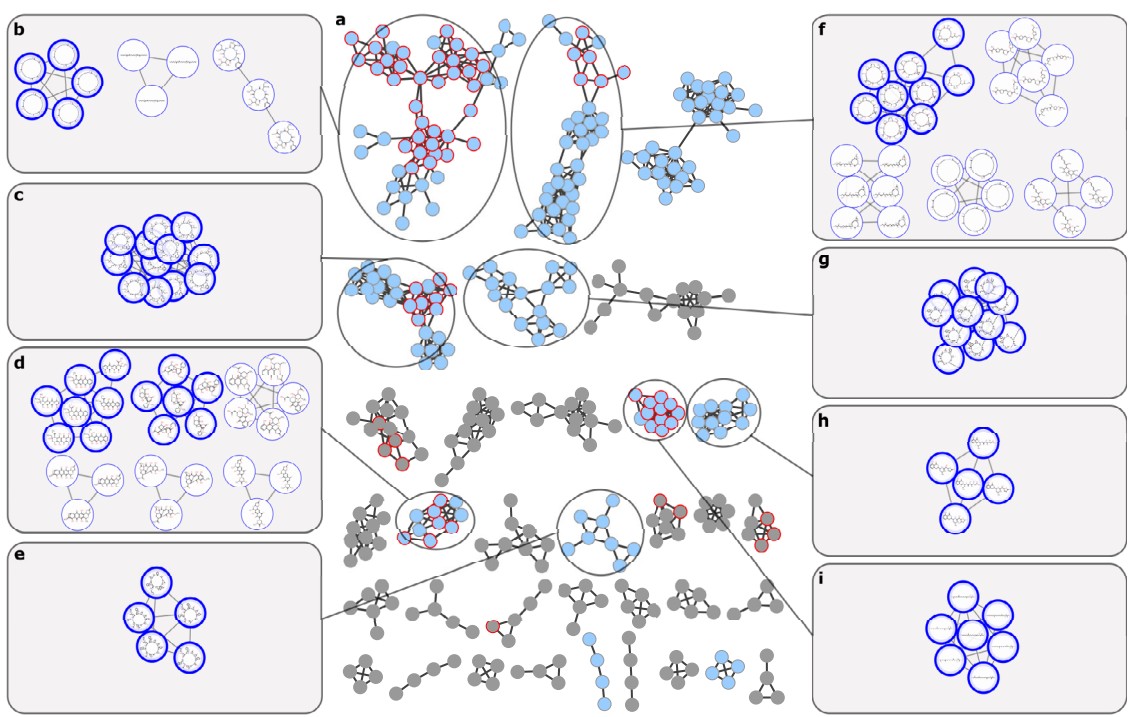

**Fig. 5 | Bacterial molecular network with SNAP-MS annotations. a** Molecular network for 925 marine bacteria prefractions, filtered to remove solvent and media blanks and subnetworks with fewer than four nodes. Nodes annotated by GNPS spectral libraries are outlined in red. Subnetworks annotated by SNAP-MS are colored blue. **b–i** SNAP-MS compound family annotations for select subnetworks. Nodes with bold outlines are from the top-ranked compound families. For full spectral characterization of each compound family see Supplementary Figures 5–11.

house marine bacteria extract library (Supplementary Note 1, Methods). Annotating subnetworks from complex mixtures is significantly more challenging than subnetworks from pure compound libraries because, unlike the network in Fig. 4 where each compound is represented by nodes of common adducts, subnetworks from extract libraries can include multiple nodes for a single molecule (e.g., adducts, fragments, multiply charged species) that increase the number of candidate structures. After filtering to remove artifacts and media components (Methods) the molecular network contained 34 subnetworks containing four or more nodes (Fig. 5). SNAP-MS annotation afforded compound family annotations for 11 of these subnetworks, more than half of which were not annotated using GNPS reference libraries. Overall, six subnetworks were annotated only by SNAP-MS, five by both SNAP-MS and GNPS, and four only by GNPS.

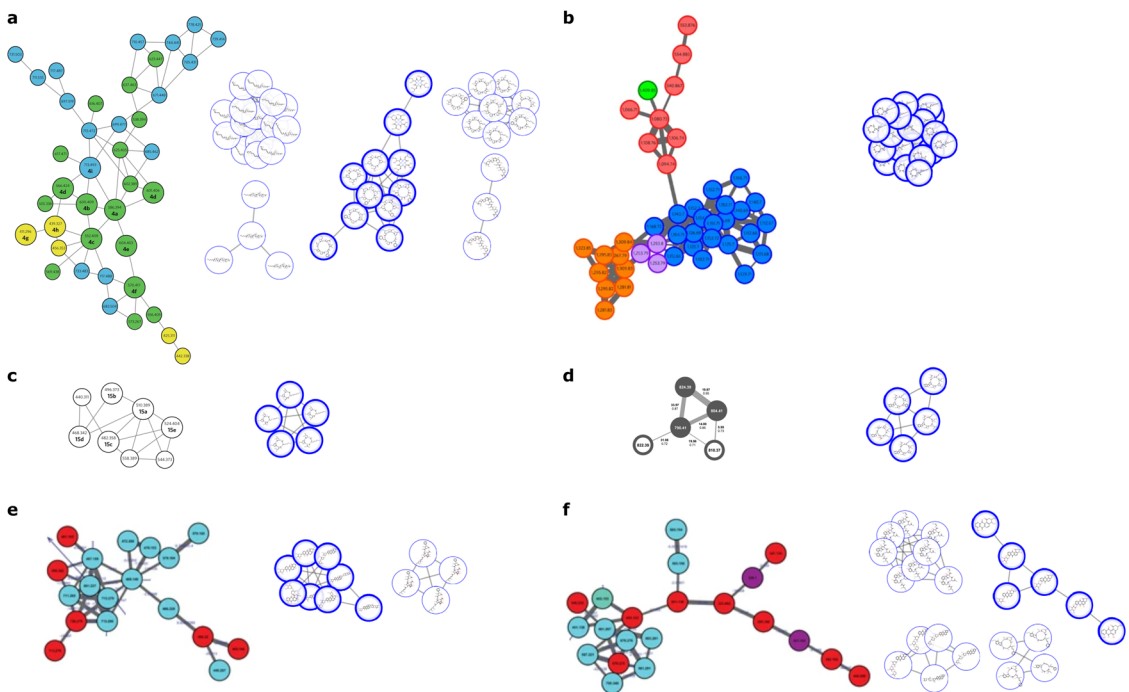

**Fig. 6 | SNAP-MS analysis on published GNPS subnetworks.** Reproductions of subnetworks from recently published data and their corresponding compound family annotations generated by SNAP-MS for **a** GameXPeptides **b** cyclic lipopeptides **c** xefoampeptides **d** noursamycins **e**, **f** angucyclines. Figure adapted from refs. [14–17] with permission. Node colors derive from color codings in original publications.

Encouragingly, all five of the subnetworks annotated by both platforms gave complimentary annotations in each case (Fig. 5b–d, f, i). To validate the predictions made by SNAP-MS we investigated example fractions from each subnetwork using a combination of isolation/ NMR analysis, MS[2] data, and co-injections (Supplementary Note 2). Of these, seven contained sufficient material for unambiguous structure identification, leading to confirmation of the predictions for the desferrioxamines (Fig. 5b, Supplementary Fig. 5), surugamides (Fig. 5c, Supplementary Fig. 6), enterocins (Fig. 5d, Supplementary Fig. 8), CDAs (Fig. 5e, Supplementary Fig. 9), nonactins (Fig. 5f, Supplementary Fig. 11), mycosubtilins (Fig. 5g, Supplementary Fig. 7), and amicoumacins (Fig. 5h, Supplementary Fig. 10). In two cases we could not obtain sufficient material for unambiguous assignments, while in a further two cases inspection of the original MS data indicated that the molecular networking subclusters were comprised exclusively of fragments of larger parent molecules, preventing accurate annotation by SNAP-MS. For a full description of verified subnetworks and associated physical data see Supplementary Information (Supplementary Figs. 5–11).

Finally, we reviewed the data for the four cases that were only annotated by GNPS. Two of these corresponded to compound families with no precedent as bacterial metabolites; likely false positives from the GNPS MS[2] spectral matching algorithm. The other two cases corresponded to putatively new analogues of natural products predicted using the analog search parameter in GNPS. SNAP-MS requires a reference database of known chemical structures, precluding predictions for subnetworks that only contain novel chemical structures. Therefore, combination of the GNPS and SNAP-MS annotation tools offers a powerful complement to characterize both the known and unknown molecular space of extract libraries.

## Identification of compound families from published molecular networking data

It is well recognized that both instrument hardware configuration and experiment parameters can significantly affect the composition of MS[2] spectra[13]. In principle, differences in MS[2] spectra could change the composition of subnetworks, reducing the effectiveness of the SNAP-MS platform for compound family annotation. To test the SNAP-MS platform under a range of acquisition conditions we analyzed six subnetworks from recent publications describing compound identification using molecular networking (Fig. 6). These included subnetworks containing GameXPeptides[17] (Fig. 6a), cyclic lipopeptides[18] (Fig. 6b), xefoampeptides[17] (Fig. 6c), noursamycins[19] (Fig. 6d), and two subnetworks found to contain angucyclines[20] (Fig. 6e, f). Gratifyingly, the top ranked compound family from SNAP-MS matched the published compound family in all six cases. Importantly, the identities of all six of these compound families were proven using orthogonal analytical methods by the original authors, confirming these assignments. Another important point is that four of the compound families in these subnetworks were previously known. Therefore, in these cases the SNAP-MS platform is not merely matching masses to newly published structures in the Natural Products Atlas database, but rather would have correctly predicted the compound families if used at the time of the original studies.

## Evaluation of algorithm performance with the COCONUT reference database

One limitation of the Natural Products Atlas is that it is restricted to microbially-derived natural products, precluding annotation of molecules from other sources. In addition, it is a relatively small database (29,006 compounds), raising the concern that annotation success of SNAP-MS is driven by the limited number of candidate compounds for each mass in the reference database rather than the prioritization method itself. To address these issues we repeated the analysis of NIH Natural Products Libraries 1 and 2 using the COCONUT natural products database (Supplementary Table 2); a freely available natural products database containing over 400,000 compounds from all source organism types[16]. A total of 184 subnetworks were annotated by SNAP-MS, of which 164 were classified as correct annotations (89% accuracy). In addition, 110 subnetworks did not receive annotations, of

which just 16 were considered true negatives (i.e., the subnetwork did not contain at least three compounds with different molecular formulae that are present in the COCONUT database). The remaining 94 subnetworks for which SNAP-MS did not return an annotation were mostly due to the results graphs being too large to provide useful annotations, which is a standard filtering option in the SNAP-MS workflow. From these results we conclude that SNAP-MS provides comparable true positive rates between the Natural Products Atlas and COCONUT reference databases (90% vs. 89%), but that the false negative rate increased significantly with the larger COCONUT reference database. For users this means that the number of subnetworks for which annotations are provided may be lower when using the larger reference database, but that the accuracy for subnetworks that do receive annotations will be similar using either approach.

## Comparison of annotation accuracy between SNAP-MS and Network Annotation Propagation

Among the open-source tools that provide annotations of mass spectrometry data, Network Annotation Propagation (NAP)[9] is the most similar in function to SNAP-MS. In 'consensus' mode NAP annotates subnetworks from spectral networking using two consecutive steps. Firstly $MS^2$ spectra from each node are compared against a reference database of predicted $MS^2$ spectra using MetFrag[21] to create a list of candidate matches for each node. In the second step the order of these candidate matches is then refined using the MetFusion tool to prioritize matches within the subnetwork with higher chemical similarities to one another. The results from the SNAP-MS and NAP are therefore different, with SNAP-MS providing compound family predictions at the subnetwork level based on mass filtering and structural similarities, while NAP provides predictions at the node level, based on $MS^2$ spectral matching and refinement with compound structure similarities.

Analysis of NIH Natural Products Libraries 1 and 2 using NAP with the COCONUT database returned annotations for 3,006 nodes, comprising 294 subnetworks (Supplementary Note 3). 66% of nodes contained the correct structure within the top 10 candidate annotations, 57% within the top 5 and 30% as the top answer. At the subnetwork level we assessed annotation accuracy by considering the percentage of nodes in the subnetwork for which the correct structure was within the top candidates. We considered two different scenarios for top candidate selection; either the top 10 annotations, or only annotations from that list with high consensus scores to the top annotation (≥0.9) (Supplementary Table 3). Under the least stringent measure (correct answer in the top 10 annotations for at least 33% of nodes in the subnetwork) NAP returned 229 true positives and 63 false positives. By contrast, under the most stringent measure (correct answer similar to top annotation for at least 50% of nodes in the subnetwork) NAP returned 183 true positives and 109 false positives. While NAP was successful at capturing the correct structure within the list of candidates in many cases, it was rarely true that the correct structure (or a closely related congener) was ranked as the top annotation consistently within subnetworks, making it difficult for end users to determine which compound family a given subnetwork derives from.

As discussed above, analysis of NIH Natural Products Libraries 1 and 2 with SNAP-MS using the COCONUT reference database returned 164 true positive annotations, and 20 false positives but did not provide annotations for 110 subnetworks. Therefore, while NAP provides annotations for a larger number of subnetworks (184 vs. 292), SNAP-MS provides higher accuracy for annotated subnetworks (89% vs. 63%), particularly under the more stringent criteria that are representative of real-world applications of these tools.

## SNAP-MS is available as an open-access online tool

SNAP-MS is freely available to the research community via the Natural Products Atlas website (https://www.npatlas.org/discover/snapms/) (Supplementary Figs. 12, 13). The webpage includes a simple drag-and-drop interface that accepts either a GNPS network file or a list of *m/z* features. Therefore, users can either annotate full molecular networks, or submit lists of related masses derived from other processing platforms (e.g., XCMS[22] or MZmine[23]). The reference database can be defined from the Natural Products Atlas database at any taxonomic rank. For example, it is possible to filter by phylum (e.g., Cyanobacteria) or genus (e.g., *Streptomyces*) as appropriate for the sample set under investigation. This is important as reference database selection has a significant impact on annotation accuracy (Discussion). Alternatively, users may select the COCONUT reference database, which contains 406,920 natural products from a wide range of source organism types (plants, microorganisms, marine invertebrates etc.). Finally, users have the option to change default parameters (e.g., *m/z* ppm error) before submitting the job. The output files for SNAP-MS are a Cytoscape[24] file containing a collection of rank-ordered compound families for every annotated subnetwork and a folder containing GraphML files for each collection of rank ordered compound families.

## Discussion

Compound grouping by $MS^2$ spectral networking is now widely employed in natural products research. However, identification of these subnetworks remains challenging because of the limited availability of authentic reference spectra. For example, of the 29,006 compounds in the Natural Products Atlas only 1243 currently have links to reference spectra in GNPS. Tools such as DEREPLICATOR + that predict $MS^2$ spectra can improve annotation rates but are biased towards compound classes such as peptides that have reliable and predictable gas phase fragmentation mechanisms[25]. SNAP-MS annotations are independent of compound class and require no external data beyond inclusion of reference compound structures in the Natural Products Atlas.

Selecting the appropriate reference database is an important consideration for the SNAP-MS method. Previous studies have demonstrated that structural overlap between source organism types (e.g., marine macroalgae vs. cyanobacteria) is low[26]. The inclusion of reference compounds not related to the source organism(s) under investigation increases the number of candidate compounds for each mass in the subnetwork and raises the possibility of erroneous compound family assignments. For example, the analysis against our marine bacterial library discussed above employed a reference library containing two relevant phyla (Actinobacteria and Firmicutes; 7175 total compounds) to yield 11 subnetwork predictions. Rerunning the analysis with a reference library containing all bacterial compounds (11,264 total compounds) returns one additional prediction (Supplementary Fig. 14). However, this prediction was for a compound family that is widely distributed among cyanobacteria but has never been found in Actinobacteria or Firmicutes; likely a false-positive annotation. Similarly, replacing the Natural Products Atlas reference database with the much larger COCONUT database reduced the number of subnetworks for which confident annotations could be provided. SNAP-MS is the only tool for annotating molecular networks that permits filtering by taxonomic rank. It is strongly recommended that, where possible, users make use of this feature to improve prediction accuracy.

It is important to stress that this approach provides compound family annotations for subnetworks, not definitive identifications of individual nodes. The platform identifies compound families containing appropriate molecular formulae but does not consider structural features of individual members or the $MS^2$ data for individual nodes. Because many compound families include isobaric members and many natural products samples include novel compounds it is not possible to use this approach for individual compound identification, or to identify 'singleton' masses that are not part of larger subnetworks. Instead, SNAP-MS is designed to 'dereplicate' subnetworks containing

known compound families to inform and accelerate downstream isolation and identification work.

A number of other tools exist for predicting compound classes from MS[2] spectra[12,27,28]. Of these, the CANOPUS tool[29] that is part of the SIRIUS data analysis package[30] is quickly becoming adopted by members of the natural products community. CANOPUS predicts ChemOnt ontological classifications for unknown analytes by extracting molecular features from experimental MS[2] spectra and comparing these feature fingerprints to those of over 4 million known compounds that have been annotated by ClassyFire[31]. All entries in the Natural Products Atlas have also been annotated using ClassyFire, providing an opportunity to directly compare the annotation information offered by both platforms.

Although CANOPUS contains 2497 ClassyFire classes, many of these are not relevant to natural products and only 223 are present in the Natural Products Atlas. By contrast, the Natural Products Atlas contains 1901 compound families (excluding families with fewer than three members) based on structure similarity grouping. Among these families, 1211 contain consistent ClassyFire class assignments, with the remaining 690 containing multiple ClassyFire classes. Therefore, SNAP-MS provides a valuable complement to the data obtained from the CANOPUS package by offering fine-scale family annotations that subdivide the broader ontological classifications provided by CANOPUS.

Despite the success of the SNAP-MS approach, a number of potential future directions exist that would further improve the accuracy of the methods. Integration of tools such as SIRIUS[30] that can accurately derive molecular formulae from MS data would improve cluster annotations by eliminating the need to search the Natural Products Atlas for a range of possible adducts ([M + H]+, [M + Na]+ etc.) for each mass, reducing the number of candidate molecules for each subnetwork. It is tempting to suggest that comparison of MS[2] spectra for each node against predicted MS[2] spectra for candidate annotations could be used to further improve annotation ranking. To test this hypothesis we generated cosine scores between experimental and predicted MS[2] spectra for selected subnetworks from this study (Supplementary Note 4). In most cases, cosine scores were moderate to weak, with DDA MS[2] spectra outperforming DIA spectra. In most cases cosine scores did not improve annotation accuracy, suggesting that further development in MS[2] spectral prediction is needed before this approach becomes practical for large-scale library annotation.

Optimizing the chemical clustering method for alignment with newer MS clustering tools such as Spec2Vec[32] is another attractive development that would reduce situations where compound families become incorrectly fragmented into smaller subnetworks due to poor MS[2] spectral networking. This would have the added advantage of reducing the frequency of 'super-clusters' made up of several different compound families that have been incorrectly connected due to poor spectral networking. Use of tools such as ion identity molecular networking[33] would also improve accuracy by reducing the complexity of subnetworks being annotated, and preventing erroneous annotations of subnetworks composed of many fragments of the same compound. Finally, further improvements to the Natural Products Atlas including addition of missing compounds from the historical literature and inclusion of all instances of discovery, rather than just the original isolation would improve reference database coverage, with a consequent improvement in annotation accuracy.

In conclusion, SNAP-MS offers a unique mechanism for the annotation of compound families from MS-based feature grouping approaches such as molecular networking. The method was successful at annotating subnetworks possessing greater that three nodes from pure compound reference libraries (coverage of 59% using the COCONUT database and 77% for NPAtlas database relevant subnetworks), in-house extract libraries (coverage of 26%), and published molecular networks, with an overall accuracy of 89% (194 correct top-

ranked results out of 219 annotated subnetworks). SNAP-MS is available as an open access online tool for the automated dereplication of molecular network graphs or lists of related *m/z* features, making it appropriate for integration to a wide range of different microbial natural product discovery workflows.

## Methods
### Molecular networking
The current study was conducted using three LC-MS datasets. The first, named "NIH Natural Products Library Round 1" contains 1,267 spectra and is provided on GNPS[11] (https://gnps-external.ucsd.edu/gnpslibrary/GNPS-NIH-NATURALPRODUCTSLIBRARY.json), the second, named "NIH Natural Products Library Round 2" contains 7,915 spectra and is provided on GNPS (https://gnps-external.ucsd.edu/gnpslibrary/GNPS-NIH-NATURALPRODUCTSLIBRARY_ROUND2_POSITIVE.json), and the third, the actinobacterial dataset, derived from in-house mass spectrometry data (https://massive.ucsd.edu, MSV000089680).

A molecular network was created from the first dataset using the online workflow (https://ccms-ucsd.github.io/GNPSDocumentation/) on the GNPS website (http://gnps.ucsd.edu). The data were filtered by removing all MS/MS fragment ions within +/− 17 Da of the precursor m/z. MS/MS spectra were window filtered by choosing only the top 6 fragment ions in the +/− 50 Da window throughout the spectrum. The precursor ion mass tolerance was set to 2.0 Da and a MS/MS fragment ion tolerance of 0.5 Da. A network was then created where edges were filtered to have a cosine score above 0.7 and more than 6 matched peaks. Further, edges between two nodes were kept in the network if and only if each of the nodes appeared in each other's respective top 10 most similar nodes. Finally, the maximum size of a molecular family was set to 100, and the lowest scoring edges were removed from molecular families until the molecular family size was below this threshold. The spectra in the network were then searched against GNPS spectral libraries. The library spectra were filtered in the same manner as the input data. All matches kept between network spectra and library spectra were required to have a score above 0.7 and at least 6 matched peaks. A molecular network was built using a combination of the spectra from both the NIH Natural Products Library Round 1 and Round 2 using the same workflow described above. A molecular network was generated for the actinobacterial dataset using the same parameters, except that mass tolerances were set to 0.02 Da for precursor ions and 0.02 Da for fragment ions.

The three molecular networks are available via GNPS at https://gnps.ucsd.edu//ProteoSAFe/status.jsp?task=868a61e685cb401385f1c24bc0edbe62 (NIH 1), https://gnps.ucsd.edu/ProteoSAFe/status.jsp?task=d909a4dccc2747218f9a290d05e7841a (NIH 1 and 2), and https://gnps.ucsd.edu/ProteoSAFe/status.jsp?task=2c39751a78824609a0fdadad989003b6 (actinobacterial library).

### Structural similarity score selection
For SNAP-MS to correctly identify the compound families generated by MS-based molecular networking there must be agreement between the composition of compound families generated by cheminformatic chemical fingerprinting and by MS-based molecular networking. To select an appropriate chemical fingerprinting method, we built structural similarity networks in which nodes represent molecules that are connected by edges if they were above a set similarity score cutoff. Structural similarity networks were built using the molecules present in the "NIH Natural Products Library Round 1" LC-MS dataset and edges were created using Morgan's, Molecular ACCess System (MACCS) or atom-pair fingerprinting with either Sørensen–Dice coefficient or Tanimoto coefficient at scores covering the range of 0.01 to 1.00. Each structural similarity network was compared to the MS-based molecular network built using the "NIH Natural Products Library Round 1" to determine the percentage of correctly created and correctly excluded

edges. The most appropriate structural similarity networking method was selected by choosing the method with the highest percentage of correctly created edges when 99.5% of edges were correctly excluded. We set the 99.5% threshold to minimize the creation of large compound family 'super-clusters' that would provide little to no useful information for annotation.

### Structural similarity Network Annotation Platform for Mass Spectrometry (SNAP-MS) workflow

SNAP-MS accepts input data in the form of comma-separated values (CSV) with a list of parent masses or as a GraphML network file, available from the GNPS results pages. Given a GNPS network file that contains a number of subnetworks, each subnetwork is converted into a separate mass list and filtered based on selected parameters (minimum/maximum GNPS cluster size). From this point on, each input follows the same workflow. For a given mass list, masses are binned into compound groups within a user-defined ppm window, with a single representative mass from each compound group being used going forward. Each representative mass is searched against the selected database using user-selected taxonomic grouping (NPAtlas), adducts and fragments, and the defined ppm window. Users are encouraged to use the taxonomic rank filtering option to remove candidate compound families with no relevance to the sample set (e.g., fungal compounds for cyanobacterial samples). Structural similarities are calculated for all candidate structures between compound groups using Morgan fingerprinting (radius = 2) and Dice similarity scoring (0.71 cutoff) to create compound families. Compound families are filtered to remove those families which have fewer members than the user-defined "minimum NP Atlas annotation cluster size". Occasionally the results graphs contain very large numbers of nodes, limiting their use for compound family identification and significantly impacting the performance of both the webserver and the Cytoscape desktop software. Graphs are therefore filtered by Maximum Node Count (default = 2000) and Maximum Edge Count (default = 10,000) to remove these very large graphs. Both of these variables can be adjusted in the parameters section of the webpage. Filtered results are returned as a network file in which each mass list has its own results graph. Candidate structures are presented as nodes, with edges between nodes if the two structures share a Dice similarity score above the defined threshold. This results file can be downloaded as either a zipped folder of individual graphML networks, or a single Cytoscape session file (.cys) which can be visualized using the network analysis software Cytoscape[24] (http://cytoscape.org/). SNAP-MS is available at www.npatlas.org/discover/snapms. Full documentation for the platform can be found at https://liningtonlab.github.io/snapms_documentation/.

### Generating compound family predictions

Compound family predictions were generated for the combined NIH Natural products Library dataset, the marine bacterial dataset, and the six subnetworks from published datasets[17–20] using the SNAP-MS platform. Appropriate parameters for each SNAP-MS analysis were selected based on the dataset used and can be found in the supplementary information (Supplementary Table 4).

### Validation of compound family identifications

Compound family predictions made by SNAP-MS in the actinobacterial dataset were validated through a combination of isolation and NMR analysis, or co-injections against reference standards. The predictions of the desferrioxamine and surugamide compound families were confirmed by isolation of desferrioxamine E and surugamide A from prefractions RLUS-2153C and RLUS-2144D, respectively (Supplementary Figs. 5 and 6) The predictions of the amicoumacin and myco-subtilin/iturin compound families were confirmed by isolation of AI-77-B and mycosubtilin D from prefractions RLUS-2079C and RLUS-2090B, respectively (Supplementary Figs. 10 and 7). The predictions for the

enterocin and CDA compound families were confirmed through co-injection of prefractions RLUS-2153D and RLUS-2052C with reference standards (Supplementary Figs. 8 and 9). Finally, the prediction for the nactins compound family was confirmed by the isolation of a nactin analogue from prefraction RLUS-2210D (Supplementary Fig. 11). Experimental details are available in Supplementary Note 2.

### Reporting summary

Further information on research design is available in the Nature Portfolio Reporting Summary linked to this article.

## Data availability

The three molecular networks generated in this study have been deposited in the GNPS database are available at https://gnps.ucsd.edu//ProteoSAFe/status.jsp?task=868a61e685cb401385f1c24bc0edbe62 (NIH 1), https://gnps.ucsd.edu/ProteoSAFe/status.jsp?task=d909a4dccc2747218f9a290d05e7841a (NIH 1 and 2), and https://gnps.ucsd.edu/ProteoSAFe/status.jsp?task=2c39751a78824609a0fdadad989003b6 (actinobacterial library). The two MS reference libraries used in this study are "NIH Natural Products Library Round 1" containing 1,267 spectra available on GNPS (https://gnps-external.ucsd.edu/gnpslibrary/GNPS-NIH-NATURALPRODUCTSLIBRARY.json), and "NIH Natural Products Library Round 2" containing 7,915 spectra and available on GNPS (https://gnps-external.ucsd.edu/gnpslibrary/GNPS-NIH-NATURALPRODUCTSLIBRARY_ROUND2_POSITIVE.json), The actinobacterial mass spectrometry dataset has been deposited in the massIVE database (https://massive.ucsd.edu) under accession code MSV000089680. The Natural Product Atlas database used in this study was v2020_06 and is available from Zenodo (www.zenodo.org) under https://doi.org/10.5281/zenodo.6783958. The COCONUT database used in this study was version January 2022 (https://coconut.naturalproducts.net/download).

## Code availability

SNAP-MS is freely available at www.npatlas.org/discover/snapms. All code for the SNAP-MS platform is available on GitHub (https://github.com/liningtonlab/snapms) under an MIT license. Documentation for the platform is available at https://liningtonlab.github.io/snapms_documentation/. Version 1.2.0 is archived at Zenodo under https://doi.org/10.5281/zenodo.7396660.

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

## Acknowledgements

Funding for this project was provided by the National Institutes of Health (AT008718 to RGL), NSERC Discovery grants (R.G.L. and C.A.G.) and the CREATE Training Program in BioActives (510963-2018, C.A.G and N.J.M.). We thank Scott D. Taylor for providing authentic standards of CDA3A and CDA4A and K. Kurita for isolation of trichostatin A.

## Author contributions

R.G.L and N.J.M. designed the project and created the SNAP-MS algorithm. J.A.v.S. designed and built the webserver. N.J.M., R.G.L., and C.A.G. analyzed the results. T.N.C., N.J.M., J.F.P.H., and E.J.M. generated mass spectrometry data and performed compound identifications. All authors contributed to and approved the final manuscript.

## Competing interests

The authors declare no competing interests.
