## [Peer Review File · Nature Communications]

REVIEWER COMMENTS

Reviewer #1 (Remarks to the Author):

This paper tackles one of the key problems in the field of natural product annotation. The authors just developed a tool named SNAP-MS, which can be used to annotate grouped features from the same structure similarly subnetwork (molecular network).

A lot of tools/algorithms have been developed for compound annotation utilizing the structure similarly network (molecular network). In this manuscript, they utilize the Natural Products Atlas database, and the hypothesis that compounds with the same formulae may be in the same compound family. This is interesting and can benefit all the metabolomics filed not only the natural product field.

The manuscript is written well and most of the text is easy to read and understand.

I have the following specific comments:

1. Most of the details for the algorithm/tool are not clear in the main text which makes sense, this is why the method section is very necessary but unfortunately I can't find it in both the manuscript and supplementary. Please check it.
2. The authors have demonstrated that the compounds' formulae can be used to "predict" their compound families. However, I think it will be more convincible that the author can give confidence that one formula belongs to one compound family.
3. Because I didn't find the compound family detail in the text, so I guess it should be the "classes" for compounds. We know that compound classes have different "levels" (superfamily, subfamily), so I am wondering which level family they used to demonstrate their hypothesis in Fig. 1.
4. The SNAP-MS just uses the formula-compound family to remove the redundant annotations for each feature based on accurate mass. However, we know that for the features, there is a lot of information we can use, such as adducts, isotopes, biological knowledge, just like other tools shows. So, I am wondering if the authors have tried to integrate this information into SNAP-MS and if this can increase the annotation accuracy?
5. As a methodology (new tool, method) paper, the authors should compare their tools with other published tools and show the advantages of SNA-MS.
6. Not a big deal, but I have to say that most of the figures are not so clear for readers to understand, the authors should consider redesigning their figures to make them more informative.

Reviewer #3 (Remarks to the Author):

The manuscript describes a method for annotating molecular families or subnetworks by searching in a structure database for structural related candidates. This idea is not entirely new and similar network methods were published recently, including:

- Rogers et al. 2009 in Bioinformatics, who was using Gibbs sampling on networks to identify molecular formulas
- Chen et al. 2021 in Nature Methods, who was using an ILP on networks for molecular formula identification
- da Silva et al. 2018 in PLOS, who used network methods to enhance molecular structure annotations
- Rasche et al. 2012, Treutler et al. 2016, and Ernst et al. 2019 who used clustering and network methods to assign compound classes to molecular subnetworks

The authors did not cited any of these methods.

In particular the "Network Propagation Annotation" method by da Silva can be seen as a more advanced variant of the method described here.

All these network methods come with the same limitations and problems: They work great as long as the structure database covers most of the measured metabolites. This is rarely the case in real world studies, as most metabolites (in particular in non-model organisms) are not contained in the structure databases. Obviously, if I choose a database small enough, mapping masses to unique structure candidates becomes simple. Network methods often show a very good performance in evaluations, because these evaluations are carried out on reference data (where database coverage is very high). The same happens here: the GNPS NIH collection consists of natural products standards, most of them are very likely also part of the natural product atlas (in particular, because the authors from the natural product atlas worked together with the GNPS community to create crosslinks between both platforms). The authors even write "We selected these reference libraries because they were publicly available, contained MS2 data for individual pure compounds, and overlapped with compounds in the Natural Products Atlas", or in other words: we used a library for which we know that our method will work. The second evaluation set are measurements from marine bacteria extracts, again taxons that are well represented in the natural product atlas (in particular, because the last author from the atlas is also the last author of this manuscript). Now what happens if I run the method on a taxon which is not covered that well in the natural product atlas? In best case, I would get no answers at all. More likely is that I get a lot of spurious answers just by randomly matching masses in the atlas. When using larger databases than the Natural Product Atlas I will probably get much more spurious annotations. It seems from the manuscript that the only filter implemented is that a molecular subnetwork has to match against at least three

structural related molecules. For me this is not very convincing: there are many mass deltas belonging to "boring biotransformations" that can be found in any compound class. Take for example hydroxylation/dehydroxylation: Having a subnetwork containing delta m/z of +/-17 (+/-OH) and +/-34 (+/-H₂O₂) is a very common thing. Also, many structures in databases have hydroxylated or dehydroxylated variants. If the method finds three database hits of structures that differ in OH and H₂O₂ then the whole subnetwork is annotated by this structural cluster. I would assume that there will be a LOT of wrong annotations when using a larger structure database. So the reason why this method works so well in evaluations might be the size of the database: the natural product atlas is one of the smallest (but best curated) structure databases. With 29,000 compounds it is twelve times smaller than the libraries used in the "Network Propagation Annotation" manuscript. There are almost no isobaric compounds contained in the atlas, so each exact mass hit is already a unambiguous molecular formula annotation. In fact, due to the small size of the library one would probably already get similarly good results in the evaluations by just searching the m/z in the library; see also Figure 1 that shows that a single molecular formula assignment is an unique mapping to a compound cluster in 78% of the cases.

So all in all the method has two problems: first, it only works for the few compounds which are part of the natural product atlas. So instead of spectral library search, which covers only around 30,000 compounds, we now search in structure databases that are also just containing 30,000 compounds. I do not expect a big gain in the number of annotations (in the second evaluation, the number of putative identified molecular families was increased by factor two). Second, it is unclear and not evaluated in the manuscript how often the method fails and annotates metabolite families with spurious structure hits. In the marine bacterial extract, a small database of 7,000 structures from marine bacteria were used which ends up in 11 annotated metabolite families. By adding 4,000 additional structures of other bacteria, the authors got one additional (probably) wrong annotation. 1 of 12 is an error rate of 8%. But this is still using a tiny database of 11,000 structures. Using the complete natural product atlas will probably results in a larger error. Using a larger database like COCONUT with 400,000 compounds will probably end up in many wrong annotations. Now the authors might argue that their method should be used in cases where I have a structure database exactly for the taxon I'm interested in. But this works only if the taxon is well researched and most of the structures I might expect for this taxon are part of my database. Any method that expands my structure database (for example by using biotransformations, as it is done in the MINES or in Biotransformer) will blow up the method.

In my opinion such a method is only of limited use: it can only be used in the few cases where I already know most metabolites in my samples. However, in these cases I could simply do an m/z matching against my database and I would get similarly good results.

Furthermore, there are already network methods that may have similar problems, but at least work on much larger structure databases: Network Propagation Annotation is annotating structures using network information, but it also utilizes the MS/MS (which is just "thrown away" in the method

presented). Note that MS/MS is also necessary for the SNAP-MS method, because it is used to build the network structure. Thus, not using the MS/MS after building the network is a big disadvantage of SNAP-MS. Additionally, with MolNetEnhancer there is already a method that annotates whole molecular families instead of single compounds. The manuscript is neither citing these alternative approaches, nor is it evaluating against any competing method.

For publication the authors would have to show that:

- the method works also with large databases like COCONUT
- make a proper evaluation about the number of true positives and false positives
- evaluate against competing methods like Network Propagation Annotation and simple m/z search

origin_organism_type	original_gnps_mass	adduct	compound_name	top_candidate
Bacterium	607.1354	2m_plus_na	3-hydroxy-1,4-diphe...	[x]
Bacterium	607.1354	2m_plus_na	Gaburedin E	[x]
Fungus	607.1354	2m_plus_na	Polyporic acid	[x]
Bacterium	607.1354	2m_plus_na	Ralfuranone I	[x]
Fungus	607.1354	2m_plus_na	Volucrisporin	[x]
Fungus	607.1354	2m_plus_na	Xerulinic acid	[x]
Bacterium	293.0805	m_plus_h	3-hydroxy-1,4-diphe...	[x]
Bacterium	293.0805	m_plus_h	Gaburedin E	[x]
Fungus	293.0805	m_plus_h	Polyporic acid	[x]
Bacterium	293.0805	m_plus_h	Ralfuranone I	[x]
Fungus	293.0805	m_plus_h	Volucrisporin	[x]
Fungus	293.0805	m_plus_h	Xerulinic acid	[x]
Fungus	293.0805	m_plus_h_minus_h2o	Phenguignardic acid	[x]
Fungus	293.0805	m_plus_h_minus_h2o	Phlebiopsin A	[x]
Bacterium	315.0621	m_plus_na	3-hydroxy-1,4-diphe...	[x]
Bacterium	315.0621	m_plus_na	Gaburedin E	[x]
Fungus	315.0621	m_plus_na	Polyporic acid	[x]
Bacterium	315.0621	m_plus_na	Ralfuranone I	[x]
Fungus	315.0621	m_plus_na	Volucrisporin	[x]
Fungus	315.0621	m_plus_na	Xerulinic acid	[x]
Bacterium	356.0893	m_plus_na	(+)-pratensilin A	[ ]

Overall, SNAP-MS is a very snappy tool easily accessed through its web interface with direct ties to the Natural Product Atlas. The approach to align spectral networking and molecular fingerprint networking results and incorporating filtering options by a curated knowledge base is new and provides useful information to the enduser about molecular families, that are described by MN subnetworks with at least two or three nodes from known compounds/molecular formulas. The manuscript is well written with great figures and SI. However, there is one major and some minor concerns. As well as some ideas for future directions or rounds of revision.

Major

1. Page 5-6 ("*Molecular formula distributions are diagnostic for compound family*") and Figure 1 need rework based on the following points:
 - a. This part is the key to providing evidence that molecular formulas fall within the same molecular families and are therefore diagnostic for them. However, the provided statistics suffer from the still limited known chemical space. Most of the unique formulas (66%) of natural products have single entries in NPA. By subtracting the 8,349 single entry formulas from the 78% that were found in a single molecular family, most of the multiple entry formulas are present in multiple molecular families. The 78% as a number signals the opposite and might not be a good measure. Furthermore, calculating the same measure on pairs and triples of formulas naturally boost the number from 78% to 99.99%, because of the high chance of one or more single entry formulas in the triple.
 - b. This means that the validity of **figure 1** is questionable, as it is based on these results, which are heavily influenced by single entry formulas.
 - c. Apart from this concern with the figure, a total number of sets and triples would be good, maybe below the relative numbers. the Log (=log10?) does

not provide more insight but rather describes the problem with the single entry formulas.

- d. As a possible solution to the problem: While it does not feel right to remove all single entry formulas before doing this type of statistical analysis, as this would artificially narrow the search space, this could still provide insight into the ratio of multi entry formula triples found across different molecular families. The excerpts of this comparison described in the text (e.g., the most commonly found triple is only found in 5 molecular families) is a valid description and might be extended by the analysis of multi entry formula, if applicable.

Minor

2. The description of MN parameters and the exact workflow used on GNPS is missing. I suspect the natural product library was processed by classical MN with the library mgf as input. Links to the GNPS jobs as suggested in 3c might be a sufficient reference for the used parameters - or a list of settings.
3. Data and code availability and license information are missing in this version.
 - a. Would be great to know if the source code will be open-source?
 - b. Please list all data accession codes (e.g., linking to repository entries on MassIVE, MetaboLights, MetabolomicsWorkbench) for the external and internal studies/datasets if available.
 - c. List all GNPS jobs and link them to their datasets - this is an easy way to access the graphml files, library matches, and recreate the presented results with SNAP-MS.
4. The term **subnetwork** describes MN subnetworks, however, the manuscript describes two types of networks: Spectral networks (MN) based on cosine similarity and molecular family networks based on structural similarity.
5. The term **MS² spectral matching** should be better exchanged with **MS² spectral networking**. Matching might point to spectral library matching, which is also part of molecular networking. Networking on the other hand usually employs modification aware spectral similarity.
6. *“some instruments are not even capable of acquiring MS2 data (e.g., TOF systems)”* - We all love the data quality of FTMS instruments, which we buy with a slower acquisition rate etc, but this random “TOF systems” (TOF-MS) reference feels out of place. The rest of the statement is true and also applies to other MS.
7. Molecular networking is not a trivial term. Should be cited and described more on page 4, where it is first mentioned in the main.
8. The number of molecular families should be accompanied by a plot of their size distribution (e.g., a histogram). Size being the number of structures or compounds or formulas
9. Where is the **“Methods”** part mentioned at several locations in the text:
 - a. *“by molecular networking of 1,267 spectra from known standards (Methods)”*
 - b. *“similarity scoring methods at a range of different similarity score cutoffs (Methods)”*
 - c. *“After filtering to remove artifacts and media components (Methods)”*
10. **Conclusion:** “accuracy of 89%” should mention the XX top hits out of YY MN subnetworks (89%). This number involves heavy filtering though (different filters for

each dataset) and should be set into context with the actual coverage, as in MN subnetworks with 3+ nodes that were annotated by SNAP-MS. Assumably, this number will be relatively low which would indicate the general problem of most annotation tools to yield a low number of (confident) annotations. In no way deminishing the important advance and insights that SNAP-MS brings into molecular networks.

Ideas (maybe for future developments or reviews)

11. **Molecular family edges:** It might be interesting to look at the original MN and add another layer of edges, connecting all MN nodes that belong to the same molecular family. There is an option to add additional edges within FBMN on GNPS by defining the two connected nodes, an annotation, and score (if available). A table like this is also easy to integrate into existing networks within Cytoscape.
Example: Figure 5f; The depicted MN is rather complex and all library matches cluster together. Overlaying the new type of molecular family edges might create more density in the different parts of the MN subnetwork, proving that SNAP-MS can identify different classes within one MN subnetwork in another graphical way.
12. **Easier referencing:** the downloaded graphml/cytoscape file could also contain the original input molecular network as a reference.
13. **MN compatibility:** add GNPS MN node columns to output, especially the **cluster index** which is the unique identifier for each node in the MN (name and shared name are the same). The GNPS library matches columns, for instance, might serve as MS² derived validation for some of the structures/molecular families.
14. **Harmonizing adducts:** While there is no agreed standard notation for adducts, adaptation and integration of SNAP-MS results into other pipelines might be facilitated by something like [M+H]⁺ or M+H, which is used by most other tools. It would also boost readability.
15. **Edge scores = weight:** Edges in molecular family networks (e.g., Figure 3c) currently carry no attributes. The calculated structural similarity between two nodes would be useful to visualize networks with weighted edges (width linked to similarity). Another edge attribute might flag edges between nodes from the same input *m/z*.

We thank the review team for their detailed and thoughtful review of our manuscript, and for providing valuable recommendations to strengthen the study, which we have addressed in point form below. Overall, three key themes emerged from these reviews. Firstly, questions were raised about the accuracy of the SNAP-MS method with larger reference libraries than the comparatively small Natural Products Atlas database (29,006 compounds). To answer this question we downloaded and reformatted the COCONUT database of natural products (~400,000 compounds) and repeated the SNAP-MS analyses on the same input data using this much larger reference database. Results were broadly comparable between the two reference databases in terms of accuracy despite the large discrepancy in sizes (see point-by-point responses below), indicating that SNAP-MS can be used successfully even with a large generalist database such as COCONUT. To offer this new functionality to the community we rebuilt the webserver to include the COCONUT DB as an alternative reference database option, and reworked the data output formats to provide COCONUT compound IDs and hyperlinks in place of the Atlas versions in the results graphs if COCONUT was selected as the reference database.

Secondly, reviewer 3 highlighted Network Annotation Propagation as an alternative annotation strategy sharing some similarities with the SNAP-MS approach, and asked about the relative performance of the two methods. To assess this we performed analogous annotations of our test datasets using the NAP tool, and developed new methods to score the relative performance of the two platforms. The two platforms provide different annotation information, making direct comparisons of performance challenging. Specifically, SNAP-MS provides subnetwork level compound family predictions meaning that results are either right or wrong for a given subnetwork, while NAP provides independent predictions for the identities of every node in the subnetwork meaning that results can contain both right and wrong answers for different nodes. Despite these differences, comparison of the two approaches against the same reference dataset indicated that SNAP-MS has a significantly lower false positive rate than NAP, while NAP had higher overall coverage than SNAP-MS.

Finally, it was noted by all reviewers that details on materials and methods were lacking in the original submission. This was an oversight on our part, caused by the materials and methods being required as a separate document in the original submission to Nature Biotechnology and not being carried over during transfer to Nature Communications. We have added the methods section to the main manuscript, and augmented this with extended details in the supporting information where appropriate.

We hope that together these extensive additions will meet the standards set by the review team, and that this study will receive a favourable response.

Reviewer: 1

"This paper tackles one of the key problems in the field of natural product annotation. The authors just developed a tool named SNAP- MS, which can be used to annotate grouped features from the same structure similarly subnetwork (molecular network).

A lot of tools/algorithms have been developed for compound annotation utilizing the structure similarly network (molecular network). In this manuscript, they utilize the Natural Products Atlas database, and the hypothesis that compounds with the same formulae may be in the same compound family. This is interesting and can benefit all the metabolomics filed not only the natural product field.

The manuscript is written well and most of the text is easy to read and understand."

Action: Comment. We thank the reviewer for their positive comments about this manuscript.

- 1) *"Most of the details for the algorithm/tool are not clear in the main text which makes sense, this is why the method section is very necessary but unfortunately I can't find it in both the manuscript and supplementary. Please check it."*

Action: Addition. The methods section has now been included in the manuscript text following the discussion section. This section provides the details for the SNAP-MS workflow.

- 2) *"The authors have demonstrated that the compounds' formulae can be used to "predict" their compound families. However, I think it will be more convincible that the author can give confidence that one formula belongs to one compound family."*

Action: Clarification. It is important to stress that SNAP-MS works by exploiting the observation that sets of formulae are unique to specific compound classes, rather than individual formulae. Supplementary Figure 1a demonstrates that individual formulae can appear over 150 times in the NP Atlas dataset. However, Figure 1f shows that of all possible sets of three formulae present in compound families in the Atlas, the vast majority (97.16%) occur in just a single compound family. Therefore, it is the distribution of formulae within a compound family that is the discriminating factor for identification, rather than the presence or absence of specific formulae among the set.

- 3) *"Because I didn't find the compound family detail in the text, so I guess it should be the "classes" for compounds. We know that compound classes have different "levels" (superfamily, subfamily), so I am wondering which level family they used to demonstrate their hypothesis in Fig. 1."*

Action: Addition. To improve readability and formally define the many related terms that have proliferated in this area we have created a new glossary in the supporting information. The technical definition of "compound family" as it relates to the SNAP-MS platform is defined as

“Discreet groupings of molecules from a compound database, generated using structural similarity. Compound family constitution can vary depending on the selected chemical fingerprinting method, similarity score threshold, and chemical database.”. Functionally, this means that members of the same compound family will contain the same or very similar carbon backbones, and that decoration with other functional groups (OH, olefin, epoxide etc.) will be broadly similar among the members of the family. To browse examples of compound families, reviewers are encouraged to visit <https://www.npatlas.org/explore/clusters> select the ‘cluster id’ radio button, and enter different numbers in the cluster id box. Interesting examples include clusters 23 and 112. Hovering over each node displays the corresponding structure. The compound family term has been included in this new glossary of terms available in the SI (Supplementary Table S1).

- 4) *“The SNAP-MS just uses the formula-compound family to remove the redundant annotations for each feature based on accurate mass. However, we know that for the features, there is a lot of information we can use, such as adducts, isotopes, biological knowledge, just like other tools shows. So, I am wondering if the authors have tried to integrate this information into SNAP-MS and if this can increase the annotation accuracy?”*

Action: Clarification/ Discussion. We did consider including additional data from both MS¹ and MS² spectra to improve annotation accuracy as the author suggests. However, we also wanted to make a tool that was as easy and flexible to use as possible, and that was capable of ingesting data from the widest range of data processing packages. By limiting the input to a list of adduct masses for any set of related molecules SNAP-MS is able to directly accept graphML outputs from GNPS, while also accepting mass lists derived from any other mass spectrometry processing suite.

The point is well made that these additional data are useful and important for ranking candidate answers. We explicitly tested this idea by cosine scoring MS² data from selected subnetworks against predicted MS² spectra for all candidate annotations (Supplementary note 4). In general the results for this were poor, with even correct answers having very low cosine scores. We added a Supplementary note to the SI describing this analysis, and added a short paragraph discussing these results to the discussion section.

- 5) *“As a methodology (new tool, method) paper, the authors should compare their tools with other published tools and show the advantages of SNA-MS.”*

Action: Addition. Network Annotation Propagation (NAP) is the most similar tool to SNAP-MS, as was pointed out by reviewer 3. We have performed a new set of analyses using NAP and have compared the results from the two platforms. This work is described in a major new section in the main manuscript, and is supported by additional information in the supplementary

information. For a full discussion of this analysis please see our response to Reviewer 3, point 3 below.

- 6) *“Not a big deal, but I have to say that most of the figures are not so clear for readers to understand, the authors should consider redesigning their figures to make them more informative.”*

Action: Clarification. We found it difficult to respond to this comment given that the other reviewers were complementary about the figure design. To make the figures clearer and more informative we carefully reviewed and revised each figure caption and added additional descriptive text where required.

Reviewer: 2

“The authors describe SNAP-MS a new web tool to analyze mass spectrometry data and annotate known molecular families of structurally similar natural products. The input, a list of m/z values, is converted into a list of compounds from the Natural Product Atlas that match each m/z taking into account common adducts and a relative m/z tolerance (ppm). Based on molecular fingerprint networking, the molecular family network with the maximum number of explained input m/z is considered the top rank - but all are reported.

Here, molecular networking (MN) is used as a tool to prefilter the m/z lists by applying SNAP-MS on each MN subnetwork of ions with similar fragmentation pattern. The algorithm to establish molecular family networks was tuned to produce connections similar to those from molecular networking (I assume, based on the modified spectral cosine similarity used in GNPS). This means that the MN connections have no weight or further consequences for the results apart from defining lists of m/z values. This could be changed in the future by merging the two networks (see future ideas below). It will be interesting to see how this tool might integrate with structure predictions by SIRIUS/CSI:FingerID and other tools in the future.

The web service for SNAP-MS made it easy to run a custom analysis. Apparently, the tool can already handle graphml inputs from Ion Identity Molecular Networking (IIMN, from MZmine and GNPS) and also incorporates all connected ion identities into the m/z list. See an example below for a small IIMN (left) and the corresponding molecular family networks (right). Even the adducts seem to match in this case, at least M+H, M+Na, and 2M+Na, which were used to query NPA during SNAP-MS:

Overall, SNAP-MS is a very snappy tool easily accessed through its web interface with direct ties to the Natural Product Atlas. The approach to align spectral networking and molecular fingerprint networking results and incorporating filtering options by a curated knowledge base is new and provides useful information to the enduser about molecular families, that are described by MN subnetworks with at least two or three nodes from known compounds/molecular formulas.

The manuscript is well written with great figures and SI. However, there is one major and some minor concerns. As well as some ideas for future directions or rounds of revision."

Action: Comment. We thank the reviewer for their encouraging summary of this work. We are pleased to see that the online platform worked correctly with an externally generated graphML file, and that the assignments were in line with the expected results for this graph.

Major Concern:

- 1) *"Page 5-6 ("Molecular formula distributions are diagnostic for compound family") and Figure 1 need rework based on the following points:*
 - a. *This part is the key to providing evidence that molecular formulas fall within the same molecular families and are therefore diagnostic for them. However, the provided statistics suffer from the still limited known chemical space. Most of the unique formulas (66%) of natural products have single entries in NPA. By subtracting the 8,349 single entry formulas from the 78% that were found in a single molecular family, most of the multiple entry formulas are present in multiple molecular families. The 78% as a number signals the opposite and might not be a good measure. Furthermore, calculating the same measure on pairs and triples of formulas naturally boost the number from 78% to 99.99%, because of the high chance of one or more single entry formulas in the triple.*
 - b. *This means that the validity of figure 1 is questionable, as it is based on these results, which are heavily influenced by single entry formulas.*
 - c. *Apart from this concern with the figure, a total number of sets and triples would be good, maybe below the relative numbers. the Log (=log10?) does not provide more insight but rather describes the problem with the single entry formulas.*
 - d. *As a possible solution to the problem: While it does not feel right to remove all single entry formulas before doing this type of statistical analysis, as this would artificially narrow the search space, this could still provide insight into the ratio of multi entry formula triples found across different molecular families. The excerpts of this comparison described in the text (e.g., the most commonly found triple is only found in 5 molecular families) is a valid description and might be extended by the analysis of multi entry formula, if applicable."*

Action: Addition. The reviewer makes a valid point that the inclusion of singleton compounds will have a significant impact on the number of formula combinations that are unique, because every combination containing a singleton formula is also unique by definition. However, this is also in part why the SNAP-MS method is so successful. We agree that simply removing the singletons does not accurately represent the composition of the reference database either. To address this issue we reanalyzed the distributions of molecular formulae in the Natural Products

Atlas, removing all singleton formulae, and compared this to the original analysis. Somewhat surprisingly, removal of singleton formulae had little impact on the frequency of formula groups, with the number of unique triples decreasing by less than three percent, from 99.99% to 97.16%. Figure 1 has been updated to reflect the new analysis, and the text discussing this analysis has been updated. The original figure 1 (containing singleton formulae) has been moved to the SI (Supplementary Fig. S3).

Minor Concerns:

- 2) *“The description of MN parameters and the exact workflow used on GNPS is missing. I suspect the natural product library was processed by classical MN with the library mgf as input. Links to the GNPS jobs as suggested in 3c might be a sufficient reference for the used parameters - or a list of settings.”*

Action: Addition. The methods section has now been included in the manuscript text following the Discussion section. Molecular network parameters and links to the GNPS jobs have been included in the “Molecular networking” section of the methods and the data availability statement.

- 3) *“Data and code availability and license information are missing in this version. a. Would be great to know if the source code will be open-source?”*

Action: Addition. The codebase was not publicly available during the original submission. This has been remedied and the SNAP-MS repository is now publicly available on GitHub under an MIT license (<https://github.com/liningtonlab/snapms>). The two reference databases (NP Atlas and COCONUT) are freely available from the download sections of their respective database websites.

- 4) *“Please list all data accession codes (e.g., linking to repository entries on MassIVE, MetaboLights, MetabolomicsWorkbench) for the external and internal studies/datasets if available.”*

Action: Addition. Links to the appropriate GNPS libraries and MassIVE datasets have now been included in the “Molecular networking” section of the methods and the data availability statement. Reviewers may access the MassIVE datasets using the password, "Linington0617". These data will be made public upon acceptance of the manuscript for publication.

- 5) *“List all GNPS jobs and link them to their datasets - this is an easy way to access the graphml files, library matches, and recreate the presented results with SNAP-MS.”*

Action: Addition. Links to GNPS jobs have now been included in the “Molecular networking” section of the methods.

- 6) *“The term subnetwork describes MN subnetworks, however, the manuscript describes two types of networks: Spectral networks (MN) based on cosine similarity and molecular family networks based on structural similarity.”*

Action: Addition/ Clarification. We agree that the terminology in this study is quite complicated, and that this is made more so by closely related definitions from other studies to which we refer. To improve clarity we have created a new glossary of terms, which is available in the supplementary material (Table 1) and referred to in the main text.

- 7) *“The term MS 2 spectral matching should be better exchanged with MS 2 spectral networking. Matching might point to spectral library matching, which is also part of molecular networking. Networking on the other hand usually employs modification aware spectral similarity.”*

Action: Correction. We have updated the text to be consistent with terminology and included definitions for both terms in the new glossary in the SI.

- 8) *““some instruments are not even capable of acquiring MS2 data (e.g., TOF systems)” - We all love the data quality of FTMS instruments, which we buy with a slower acquisition rate etc, but this random “TOF systems” (TOF-MS) reference feels out of place. The rest of the statement is true and also applies to other MS.”*

Action: Correction. We have updated the text by removing the reference to “TOF systems” specifically.

- 9) *“Molecular networking is not a trivial term. Should be cited and described more on page 4, where it is first mentioned in the main.”*

Action: Addition. MS² spectral networking and molecular networking are now briefly discussed in the first paragraph on the introduction and both are included in the new glossary of terms in the SI (Supplemental table S1)

- 10) *“The number of molecular families should be accompanied by a plot of their size distribution (e.g., a histogram). Size being the number of structures or compounds or formulas”*

Action: Correction. A figure showing the number of unique chemical formulae within each compound family has been created and included in the SI (supplementary Fig. S2).

- 11) *“Where is the “Methods” part mentioned at several locations in the text:*
a. *“by molecular networking of 1,267 spectra from known standards (Methods)”*
b. *“similarity scoring methods at a range of different similarity score cutoffs (Methods)”*

c. *“After filtering to remove artifacts and media components (Methods)”*”

Action: Addition. The methods section has now been included in the main manuscript.

12) *“Conclusion: “accuracy of 89%” should mention the XX top hits out of YY MN subnetworks (89%). This number involves heavy filtering though (different filters for each dataset) and should be set into context with the actual coverage, as in MN subnetworks with 3+ nodes that were annotated by SNAP-MS. Assumably, this number will be relatively low which would indicate the general problem of most annotation tools to yield a low number of (confident) annotations. In no way deminishing the important advance and insights that SNAP-MS brings into molecular networks.”*

Action: Addition. The conclusion has been updated to include information about the percent coverage of SNAP-MS for correctly identified subnetworks in the context of the molecular network that was analyzed. Text has also been added to clarify that 89% is in reference to the number of correctly annotated subnetworks out of the total number that receive annotations (i.e. the recall).

“Ideas (maybe for future developments or reviews)”:

13) *“Molecular family edges: It might be interesting to look at the original MN and add another layer of edges, connecting all MN nodes that belong to the same molecular family. There is an option to add additional edges within FBMN on GNPS by defining the two connected nodes, an annotation, and score (if available). A table like this is also easy to integrate into existing networks within Cytoscape. Example: Figure 5f; The depicted MN is rather complex and all library matches cluster together. Overlaying the new type of molecular family edges might create more density in the different parts of the MN subnetwork, proving that SNAP-MS can identify different classes within one MN subnetwork in another graphical way.”*

Action: Discussion. This is an interesting suggestion, particularly the idea that SNAP-MS could improve subdivisions within larger subnetworks. However, this also complicates the ranking strategy because the platform would now (for example) need to differentiate between annotations that cover a lower percentage of nodes with one consistent compound class, and answers that cover a higher percentage of nodes but with two subclasses. Given this additional layer of complexity we propose to defer development of this suggestion to the next iteration of the tool.

14) *“Easier referencing: the downloaded graphml/cytoscape file could also contain the original input molecular network as a reference.”*

Action: Addition. We have added the original GNPS network to the output cys file, and have applied the standard GNPS Cytoscape styling to this graph.

15) *“MN compatibility: add GNPS MN node columns to output, especially the cluster index which is the unique identifier for each node in the MN (name and shared name are the same). The GNPS library matches columns, for instance, might serve as MS 2 derived validation for some of the structures/molecular families.”*

Action: Clarification. Applying this suggestion is not straightforward because most GNPS nodes will connect with several candidate molecules, and some candidate molecules may connect with multiple GNPS nodes (e.g. several adducts of the same compound). Furthermore we are wary about linking GNPS nodes directly to annotations because we worry that this will lead to a misunderstanding of the output and that these links will be used as identifications (which they are not) rather than compound family-level annotation of the whole subnetwork. For these practical and philosophical reasons we have elected not to add this information to the SNAP-MS output graphs.

16) *“Harmonizing adducts: While there is no agreed standard notation for adducts, adaptation and integration of SNAP-MS results into other pipelines might be facilitated by something like $[M+H]^+$ or $M+H$, which is used by most other tools. It would also boost readability.”*

Action: Correction. We have changed the adduct notation in the output graphs to follow the $[M+H]^+$ notation suggested above.

17). *“Edge scores = weight: Edges in molecular family networks (e.g., Figure 3c) currently carry no attributes. The calculated structural similarity between two nodes would be useful to visualize networks with weighted edges (width linked to similarity). Another edge attribute might flag edges between nodes from the same input m/z.”*

Action: Discussion. This is also an interesting idea. We did try experimenting with edge weighting as a function of Tanimoto similarity score in the output graphs, but found that this often made graphs quite crowded and difficult to examine without manually moving the nodes (negating the value of edge weighting). This was accentuated by the fact that edges are only retained if they meet a high threshold (>0.66), meaning that all edges already have quite high and similar values. On balance, we felt that unweighted graphs were easier to read, and that the weighted edges did not add enough new information content to justify the reduction in readability, so retained the existing unweighted edge approach.

Reviewer: 3

"The manuscript describes a method for annotating molecular families or subnetworks by searching in a structure database for structural related candidates. This idea is not entirely new and similar network methods were published recently, including:

- Rogers et al. 2009 in Bioinformatics, who was using Gibbs sampling on networks to identify molecular formulas*
 - Chen et al. 2021 in Nature Methods, who was using an ILP on networks for molecular formula identification*
 - da Silva et al. 2018 in PLOS, who used network methods to enhance molecular structure annotations*
 - Rasche et al. 2012, Treutler et al. 2016, and Ernst et al. 2019 who used clustering and network methods to assign compound classes to molecular subnetworks*
- The authors did not cited any of these methods."*

Action: Addition. We have added discussion and citations to these papers in appropriate places in the revised manuscript.

"In particular the "Network Propagation Annotation" method by da Silva can be seen as a more advanced variant of the method described here."

Action: Clarification. Reviewer 3 is correct that Network Annotation Propagation (NAP) offers an alternative strategy for annotating groups of MS features with putative compound structures. However, in contrast to SNAP-MS, which annotates an entire MS feature group with a single compound family, NAP annotates each node individually based on MS² spectral matching against an in silico database and then re-ranks these answers based on structure similarity. Therefore, the two approaches are fundamentally different in that NAP can produce both correct and incorrect annotations for a given cluster, whereas SNAP-MS is either right or wrong. Further, NAP relies on MS² spectral matching to create the initial candidate list, whereas SNAP-MS leverages compound lists directly from reference databases. If in silico MS² predictions are accurate then NAP should out-perform SNAP-MS. By contrast, if in silico MS² predictions match poorly to real-world MS² spectra then many correct candidates can be lost using the NAP method, and SNAP-MS will perform better. We have added a major new section to the manuscript where we have performed a direct comparison between SNAP-MS and NAP, as described in detail in point 3 below. This demonstrates that SNAP-MS outperforms NAP in terms of annotation accuracy, but at the expense of a higher false negative rate (i.e., SNAP-MS generates answers for a smaller number of MS clusters, but more often returns the correct answer if an answer is provided).

"All these network methods come with the same limitations and problems: They work great as long as the structure database covers most of the measured metabolites. This is rarely the case in real world studies, as most metabolites (in particular in non-model organisms) are not contained in the structure databases."

Action: Clarification. This point is well made. SNAP-MS aims to annotate known compound families in order to improve information content for MS networks. As discussed below, SNAP-MS can annotate data from classical data-dependent MS² experiments (DDA), but can also annotate mass lists from MS¹ experiments, data-independent MS² experiments (DIA) or other exotic experiment types (DIA, MSⁿ etc.). There is no requirement for either real-world or calculated MS² spectra, removing annotation bias caused by uneven data deposition (real-world spectra) or varying prediction accuracy for different compound classes (in silico spectra). While it is correct that SNAP-MS requires structures to be in the reference database it is also clear from anecdotal and experimental evidence (Pye et al, PNAS, 2017) that rates of rediscovery are increasing. Instances where users discover entirely new compound classes without any existing reference members are low. Annotating many of the known clusters using SNAP-MS will help to prioritize these rare clusters, rather than being a negative aspect of the method. For reference, only 1,243 compounds in this version of the Atlas DB (v2020_06) have associated GNPS entries, meaning that there are only real-world reference spectra for 4% of the compounds in the NP Atlas DB.

"Obviously, if I choose a database small enough, mapping masses to unique structure candidates becomes simple. Network methods often show a very good performance in evaluations, because these evaluations are carried out on reference data (where database coverage is very high). The same happens here: the GNPS NIH collection consists of natural products standards, most of them are very likely also part of the natural product atlas (in particular, because the authors from the natural product atlas worked together with the GNPS community to create crosslinks between both platforms). The authors even write "We selected these reference libraries because they were publicly available, contained MS² data for individual pure compounds, and overlapped with compounds in the Natural Products Atlas", or in other words: we used a library for which we know that our method will work."

Action: Discussion. Reviewer 3 is correct that we selected the NIH dataset for initial performance testing precisely because the correct answers were part of the reference database. One cannot assess the full performance of any platform if it is not possible for the system to obtain the correct answer (because it is missing), or if the answers are unknown (because the test material is uncharacterized). Instead, we first tested SNAP-MS against a standardized dataset of known molecules, where the accuracy of the result could be explicitly determined. An additional advantage of this dataset is that it was all acquired on the same instrument, meaning that MS² spectral similarities were expected to be excellent between members of the same compound family, and MS feature groupings were expected to be reliable, removing a significant source of confounding error. It is worth noting that this approach is commonly employed for evaluating tool performance. Indeed, the NAP study itself used a very similar strategy to test performance, filtering the test set for [M+H]⁺ ions only, and only retaining molecules that had matches to other similar structures in the test set:

"To benchmark NAP, we have created a molecular network with a subset of 5,467 MS/MS [M+H]⁺ spectra from NIST17 library that are structurally unique and have spectral similarity (cosine score ≥ 0.6) to at least one spectrum in the subset"

In the second section of the paper we repeated these analyses against a large library of extracts of unknown composition, and validated compound family predictions experimentally by either isolating and characterizing exemplars, or by co-injection with authentic standards. Finally, in the third section we analyzed publicly available experimental data from other studies that had determined structures using orthogonal methods, and tested SNAP-MS against these datasets to assess real-world performance. Therefore, we tested SNAP-MS performance in three ways: in a basic use case where the correct answer was one possible result; in a complex use case where answers were validated by isolation and NMR characterization or co-injection, and in a complex case where MS data from different researchers and different instruments were subjected to SNAP-MS analysis. In all cases, SNAP-MS performed well, yielding correct annotations in the vast majority of cases examined.

"The second evaluation set are measurements from marine bacteria extracts, again taxons that are well represented in the natural product atlas (in particular, because the last author from the atlas is also the last author of this manuscript). Now what happens if I run the method on a taxon which is not covered that well in the natural product atlas? In best case, I would get no answers at all. More likely is that I get a lot of spurious answers just by randomly matching masses in the atlas. When using larger databases than the Natural Product Atlas I will probably get much more spurious annotations. It seems from the manuscript that the only filter implemented is that a molecular subnetwork has to match against at least three structural related molecules. For me this is not very convincing: there are many mass deltas belonging to "boring biotransformations" that can be found in any compound class. Take for example hydroxylation/dehydroxylation: Having a subnetwork containing delta m/z of +/-17 (+/-OH) and +/-34 (+/-H2O2) is a very common thing. Also, many structures in databases have hydroxylated or dehydroxylated variants. If the method finds three database hits of structures that differ in OH and H2O2 then the whole subnetwork is annotated by this structural cluster. I would assume that there will be a LOT of wrong annotations when using a larger structure database. So the reason why this method works so well in evaluations might be the size of the database: the natural product atlas is one of the smallest (but best curated) structure databases. With 29,000 compounds it is twelve times smaller than the libraries used in the "Network Propagation Annotation" manuscript."

Action: Clarification. A second major addition to this revised manuscript is the addition of the COCONUT database as an alternative reference database to the Natural Products Atlas. In contrast to the NP Atlas (~29,000 compounds) the COCONUT DB contains >400,000 compounds, including compounds derived from plants and marine invertebrates as well as microorganisms. We repeated the full analysis performed with the Atlas DB using the new COCONUT DB and report the results in a new section of the paper. These results are described in detail in point 1 below. In brief, the recall (# correct annotations/ # total annotations) was very similar between NP Atlas and COCONUT (90% vs 89%), but the false negative rate (rate of absence of answer for clusters for which an answer should have been possible) increased when using the COCONUT DB. This result contradicts the assumption made by reviewer 3 above, but highlights the value of selecting a reference database that is appropriately filtered for the target organism(s) being analyzed where possible.

"There are almost no isobaric compounds contained in the atlas, so each exact mass hit is already a unambiguous molecular formula annotation."

Action: Clarification. This statement is incorrect. The version of the Atlas database used in this study (v2020_06) contains 29,006 compounds, of which only 8,349 possess unique molecular formulae within the set. Therefore, most compounds in the Atlas contain at least one other isobaric member. The mean number of compounds per formula is 2.29. Excluding singleton formulae this value is 4.79.

"In fact, due to the small size of the library one would probably already get similarly good results in the evaluations by just searching the m/z in the library; see also Figure 1 that shows that a single molecular formula assignment is a unique mapping to a compound cluster in 78% of the cases."

Action: Clarification. This statement is also incorrect. Because only 29% of compounds have unique molecular formulae, the remaining 71% of formulae would have at best a 50% probability of being selected correctly if subjected to random selection based on mass. This probability decreases significantly when considering formulae with more than two compounds (most cases) and again when accounting for mass error, or when accommodating the possibility that a given mass could be any of a large number of different adducts. SNAP-MS has a recall of 90%; far better than what one would expect by random m/z searching.

"So all in all the method has two problems: first, it only works for the few compounds which are part of the natural product atlas. So instead of spectral library search, which covers only around 30,000 compounds, we now search in structure databases that are also just containing 30,000 compounds. I do not expect a big gain in the number of annotations (in the second evaluation, the number of putative identified molecular families was increased by factor two). Second, it is unclear and not evaluated in the manuscript how often the method fails and annotates metabolite families with spurious structure hits. In the marine bacterial extract, a small database of 7,000 structures from marine bacteria were used which ends up in 11 annotated metabolite families. By adding 4,000 additional structures of other bacteria, the authors got one additional (probably) wrong annotation. 1 of 12 is an error rate of 8%. But this is still using a tiny database of 11,000 structures. Using the complete natural product atlas will probably result in a larger error. Using a larger database like COCONUT with 400,000 compounds will probably end up in many wrong annotations. Now the authors might argue that their method should be used in cases where I have a structure database exactly for the taxon I'm interested in. But this works only if the taxon is well researched and most of the structures I might expect for this taxon are part of my database. Any method that expands my structure database (for example by using biotransformations, as it is done in the MINES or in Biotransformer) will blow up the method."

Action: Discussion. Reviewer 3 is correct that this approach will only work to annotate compound families that are already well characterized. Given that the discovery of new compounds remains one of the central objectives for natural products research, annotation of all known compound families in a sample set would be of very high value to natural products researchers for prioritizing novel chemistry. Not all MS features in a given family will relate to known chemistry, but knowing to what compound family these features belong is highly valuable for project selection and prioritization. Does the community need more examples of ferrioxamine-type siderophores? Perhaps not. Without annotations it is impossible to decide

whether a given MS feature/ set of features is worth attention. By contrast, with annotations researchers can make informed, rational decisions about project selection.

Reviewer 3 suggests that spectral matching and comparison to compound databases are equivalent (~30,000 compounds each). This is not correct. Only 1,243 compounds in this version of the Atlas DB (v2020_06) have associated reference spectra in GNPS. Therefore, spectral matching is highly biased towards a small number of molecules from very well studied genera (principally *Streptomyces*, *Penicillium*, *Aspergillus* and *Lyngbya*) for which reference data have been submitted. A good example of this bias is the reporting of surugamide cyclic peptides. Not isolated until 2013, these compounds have large molecular weights and highly diagnostic MS² spectra. Since deposition in GNPS they have been 'rediscovered' many times in molecular networking papers (including this study), not because they have new and important activities, but because reference spectra now exist for this compound class. Which other unannotated compound families are hiding in plain sight in this way? It is impossible to tell using spectral matching until an example reference spectrum is deposited, and most compound families have no such reference data.

By contrast, SNAP-MS is not biased in this way, because all known structures are considered as candidates regardless of whether or not experimental data are available. It is true that annotations are limited by what is known about the chemistry from a given taxon. For example, the genus *Burkholderia* has only recently become the subject of rigorous study for new chemistry, leading to the identification of a large number of new compound classes. Here too a dereplication strategy like SNAP-MS is very valuable to natural products practitioners. If one is studying a new source of natural products and many of the compound families are not annotated by existing chemistry databases, this gives confidence that these molecules may be of high value for further investigation. As the known chemistry of this genus matures, the reference database can be updated so that newly discovered families are annotated in existing datasets. SNAP-MS is available as a Dockerized container solution for off-line use, allowing users to create custom libraries of any composition for custom studies like this. As discussed below, increasing the DB size from 29,006 to >400,000 had a negligible impact on recall for SNAP-MS (90% vs. 89%) demonstrating that the potential issue about false positives is not of concern for the SNAP-MS approach. So, using large or small reference DBs SNAP-MS provides accurate annotations for many more compound families than can be accomplished with spectral matching, returning a low false positive rate that significantly outperforms other methods (see points 1, 2 and 3 below).

"In my opinion such a method is only of limited use: it can only be used in the few cases where I already know most metabolites in my samples. However, in these cases I could simply do an m/z matching against my database and I would get similarly good results."

Action: Clarification. We respectfully disagree with this opinion. Rather than being of limited use as a simple alternative to *m/z* matching, SNAP-MS offers a new method for compound family dereplication with excellent recall. It is customizable to specific taxonomic groups, applicable to MS data acquired on almost any 'HRMS' instrument, and characterizes compound families regardless of whether experimental MS² data are available in public repositories (which is still quite rare) or whether predicted MS² spectra are of low quality (which is still quite common). For a discussion of the limitations and weaknesses of including *in silico* MS² spectra

for improving SNAP-MS performance, see supporting information note 4 and supporting information figures 15 and 16.

Further, SNAP-MS dramatically outperforms the optimal performance that can be expected from simple m/z searching (see discussion above) and outperformed molecular networking/ GNPS in real-world annotations (see figure 5). The method is supported by a simple, open access webserver, and all code required to deploy the platform locally is available for free under an MIT license. We expect that SNAP-MS will quickly become a mainstay tool for dereplication of MS data by the natural products community due to the reliability of compound family predictions compared to other available methods (see below).

"Furthermore, there are already network methods that may have similar problems, but at least work on much larger structure databases: Network Propagation Annotation is annotating structures using network information, but it also utilizes the MS/MS (which is just "thrown away" in the method presented). Note that MS/MS is also necessary for the SNAP-MS method, because it is used to build the network structure. Thus, not using the MS/MS after building the network is a big disadvantage of SNAP-MS."

Action: Clarification. SNAP-MS offers two complementary mechanisms for importing MS data for annotation. The first is a graphML file that derives from the molecular networking platform, which is in turn derived from MS² spectral matching as noted by reviewer 3. The second option is a simple mass list, which does not require MS² data but instead can be derived from any grouping method that users wish to employ. For example, a user might obtain LCMS data that includes both UV profiles and MS data for peaks in a sample and conclude, based on UV signatures, that a set of peaks are structurally related. The mass list from these peaks would be an appropriate input in the mass list panel, and does not require any MS² data. Alternatively, a user might examine a heterologous expression system for a given biosynthetic gene cluster under several different fermentation conditions, and group all the MS features generated from these experiments as a single set, which would also be an appropriate input.

Rather than being a disadvantage, the absence of a requirement for MS² data is an inherent advantage of the SNAP-MS platform, and offers a flexible, dependency-free method for compound dereplication. Users can input data from any flavor of MS experiment, including MS¹ only, data-dependent (DDA) MS², data-independent (DIA, SWATH) MS² or MSⁿ approaches. SNAP-MS is therefore inherently more useful to the natural products community than tools which require MS data of a particular format, or that conform to existing acquisition methods.

In supporting information note 4 we explicitly test *in silico* spectral matching against real-world data. In general, these predicted spectra align poorly with experimental data and decreased the accuracy of most annotations compared to native SNAP-MS annotations. Given that few real-world MS² spectra exist for natural products, and that *in silico* MS² spectra are still of variable quality we maintain that not requiring MS² data for compound family annotation is a big advantage of SNAP-MS, rather than the disadvantage that the reviewer suggests.

"Additionally, with MolNetEnhancer there is already a method that annotates whole molecular families instead of single compounds. The manuscript is neither citing these alternative approaches, nor is it evaluating against any competing method."

Action: Clarification. MolNetEnhancer and CANOPUS are both tools that can annotate clusters based on ClassyFire categories. However, although there are 2,497 classes available in this ontology, only 223 are present in the Natural Products Atlas. Therefore, these approaches offer only a coarse classification structure compared to the more detailed compound family-level classifications offered by SNAP-MS. Of the available classification tools, NAP is the platform with the closest functionality to SNAP-MS. We have expanded the manuscript substantially to include a direct performance comparison against NAP which demonstrates that SNAP-MS possesses substantially improved recall and lower false positive rate compared to the NAP method (see detailed points 1 - 3 below).

For publication the authors would have to show that:

1) *"the method works also with large databases like COCONUT"*

Action: Addition. To address this comment, we repeated all benchmarking experiments with the NIH dataset (known structures) and compared the results to those generated using the Natural Products Atlas. Encouragingly, the true positive rate (true positives/ total positives) was very similar between the two reference databases (90% for NP Atlas, 89% for COCONUT). However, the false negative rate increased significantly with the use of the COCONUT database due to an increase in the number of cases where results graphs contained too many nodes to be considered useful for annotation. This node size cutoff is a variable that users can set in the online interface. Therefore, increasing the reference database significantly (400,000 vs 29,000 compounds) had a negligible effect on the true positive rate, but did reduce the overall number of networks for which annotations were provided. This analysis is described in a new section in the main text, and is supported with new tables in the Supplementary materials.

To allow other users to take advantage of the COCONUT database we also rewrote the webserver to permit either the NP Atlas or COCONUT to be used as the reference database and updated the output file formats to provide links to either reference database. This was not requested by any of the reviewers and required a significant reworking of the underlying codebase, but provides an expanded tool for the community that now generates annotations for compounds from plants and invertebrates in addition to microbial sources.

2) *"make a proper evaluation about the number of true positives and false positives"*

Action: Addition. As discussed above we have performed significant additional analyses for this revised submission, and have defined all four terms in the confusion matrix for both SNAP-MS (Supplementary table S2) and Network Annotation Propagation (NAP; Supplementary table S3; see point 3 below). Discovery rates are now discussed in detail in several places in the text, and

are supported by detailed numerical values. Overall SNAP-MS maintained excellent true positive rates regardless of the size of the reference database, and significantly outperformed NAP in terms of true positive rate under real-world assessment criteria (89% vs. 63%; see point 3 below).

- 3) *“evaluate against competing methods like Network Propagation Annotation and simple m/z search”*

Action: Addition. We acknowledge that we did not sufficiently discuss the conceptual and functional similarities and differences between SNAP-MS and NAP in the original submission. To address this issue, we have added discussion of NAP in the introduction, and created a major new section in the main manuscript where we repeated the analysis of the NIH dataset using NAP, and assessed the relative annotation performance of the two systems. This analysis was not straightforward because the two tools work in different ways, and generate different results; SNAP-MS annotates subnetworks at the compound family level, whereas NAP annotates each node in the subnetwork separately.

Given that the central question being addressed in our study was to determine the compound family responsible for the creation of each subnetwork we asked *‘how would users determine compound family identity from NAP annotations?’*. In a perfect scenario NAP would generate a single annotation for each node, and each annotation would be correct, leading to a subnetwork where all nodes are annotated with similar structures and the determination of the compound family is straightforward. However, this expectation is unrealistic given that many very similar congeners can exist for a given molecule. Therefore, we considered two different scoring methods. In the first we asked whether the correct structure was within the top 10 annotations for either 33% (easier case) or 50% (more challenging case) of the nodes in the subnetwork. NAP also provides a consensus score for every candidate annotation, with the top annotation set to 1. Manual inspection of these candidates showed that in some cases all candidates were very similar in structure, whereas in others there were large differences between structures, and consensus scores were low. These situations are not very useful for end users, who have no mechanism to rank high or low scoring annotations other than accepting the top ranked annotation as correct. Therefore, in the second scoring method we filtered the annotation list to retain only compounds similar to the top ranked answer (consensus score ≥ 0.9) and reran the scoring method for both 33% coverage and 50% coverage as described above.

In the most stringent of these cases (consensus score ≥ 0.9 , top answer in $\geq 50\%$ of nodes) NAP generated 292 answers, of which 183 were true positives. Therefore, in a real-world scenario where users consider structural consistency of high-ranking annotations to determine compound family, NAP had a lower true positive rate (63%) than SNAP-MS (89%). However, as discussed in the new section of the manuscript, SNAP-MS has a high false negative rate when using the larger COCONUT database, meaning that SNAP-MS annotates fewer subnetworks

overall, but that those which do receive annotations are more likely to be correct than when using NAP.

Finally, we also analyzed the NAP results to determine how many nodes possessed a top answer that was structurally similar to the correct answer (Morgan fingerprint radius 2, Dice scoring, cutoff ≥ 0.9). Using this method, just 1,276 of the 3,413 annotated nodes possessed a correct structure as the top answer (37%), highlighting the challenge of interpreting NAP results in large network diagrams for average natural products researchers.

The question of mass searching is also addressed extensively above, as well as with increased numerical discussion in the formula distribution section of the revised draft.

REVIEWERS' COMMENTS

Reviewer #1 (Remarks to the Author):

The authors have addressed all my comments.

Reviewer #2 (Remarks to the Author):

Reading the comments of the other reviewers, there was quite an overlap in the parts that needed clarification or more data. In my view, the authors delivered on this point with the following:

They provide results after removing formulas that are only present in single molecular families, improving the clarity of the message in Figure 1. It is still valid to use those formulas in their method because them being singletons might indicate their unique nature. In conclusion, the results for sets of three formulas did not change.

Evaluation against a larger database (COCONUT) reveals some limitations on the annotation of large subnetworks, however, snapms provided results for the majority of subnetworks with acceptable accuracy. Although one has to consider that the tested datasets and databases are perfect matches with 100% overlap of structures. Nevertheless, real application performance will reveal itself once more people use it, which is easy due to the web interfaces, especially with COCONUT now being an option.

Those tool comparisons are always hard to fairly conduct. Still I think this is a valid point that snapms is more stringent, leading to less annotated subnetworks, while providing better accuracy.

The manuscript improved substantially and all points were addressed.